



# Critique of 'Lotka's wheel and the long arm of history': Best available historical data shows major differences pre-1970, raising new questions.

Brian Hanley[1]

[1]Butterfly Sciences, Davis, California, USA

**Correspondence:** Brian Hanley (brian.hanley@bf-sci.com)

**Abstract.**

**Background:** The true limits to economic growth and how monetary value is determined relative to energy, goods, and services, are primary unresolved questions in economics. *Lotka's wheel and the long arm of history* (Garrett et al., 2022) presents the conjecture that the past exerts an influence on the present through a ratio of $\frac{\sum_{i=0}^{n} GWP_i}{E}$, where E is energy. The

numerator of this ratio is a trailing average composed of the sum of gross world product ($GWP$) over *all* of human time, given the variable name $W$. The conjecture presents $\frac{W}{E}$ as so close to fixed that a constant $w$ is substituted for it, based on a thermodynamic argument available in past work.

The $\frac{W}{E}$ conjecture follow-on proposition is that when the first derivative of GWP, which equation includes energy, is in no-growth balance, meaning $\frac{dE}{dt} = 0$, then inflation forces real $GWP$ to zero, even when nominal $GWP$ is maintained.

*Lotka's wheel* has a 4 column supplement covering year 1-2019 CE: A. global GDP; B. $W$ (yearly results of the $\sum_{i=0}^{n} GWP_i$); C. population; and D. energy in exajoules $E$. Replicate datasets were assembled from literature for $GWP$ ($GWP_{Rep}$) and $E$ ($E_{Rep}$).

**Two problems:** First, the equation leading to the central proposition that real gross world product will become zero when $\frac{dE}{dt} = 0$ is based on erroneous use of an approximation constant for a relationship that is not actually a constant. All avenues

for support of this proposition fail. Second, prior to 1970 $\frac{W_{Rep}}{E_{Rep}}$ is radically higher than the supplement's $\frac{W}{E}$ because $GWP_{Rep}$ values are somewhat higher, and $E_{Rep}$ values are much lower. Key replicate data are high confidence.

**Conclusion:** As presented, the long arm of history hypothesis is falsified. However, the thermodynamic argument in prior work is compelling, and in 1970, a radical change in slope of $\frac{W}{E}$ occurs. From 1970 forward, the long-arm of history hypothesis as presented appears probable. I believe that prior to 1970, the long-arm of history hypothesis may be true, but that there are

other factors that are not understood at this time.

## 1  Introduction

Limits to economic growth (the life support system that humans interact with), how money is valued, and the relationship of money to energy, resources, and production, are among the most important questions in economics that remain unsolved. Tim





Garrett has published an excellent series of papers that make thermodynamic arguments regarding economic activity and the
wealth of global society (Garrett, 2011, 2012a, b, 2014, 2015). The highlights of this are about how growth occurs in economic
systems. For instance, readers with policy interests may find the concept that inflation should average over time to $\frac{1}{EROI}$ where
EROI is energy return on energy invested, to be interesting (Garrett, 2012b, p 14).

   This careful reworking of $Lotka's Wheel$ (Garrett et al., 2022) came about because I wanted to cite it as a reference in the
area of climate economics, a field that is contentious, to be mild about it. I want to be very clear about the value and difficulty
of what Tim has published in the area. The relevant aspect of Tim's work could be summarized verbally as: the capacity of a
society to make use of energy efficiently, and use that for the benefit of the society while entropy is constantly breaking down
capital produced, is what determines the real value of money. This is consistent with a recent revision to the Cobb-Douglas
equation incorporating energy (Keen et al., 2019). Tim has begun a pioneering, cross-domain effort, that is still in the "Feeling
around in a dark closet," stage of development where empirical data is missing, or may be hidden in some nook of the academy.
Keen et al. revised the Cobb-Douglas production function[1], slightly modified here with variable names more consistent with
this critique. This equation includes $E$ in the $X$ variable. Thus, $Y$ is dependent upon $E$.

$$Y = C \times (X_K)^\alpha \times K^\alpha \times L^\beta \qquad \textbf{(Energy based Cobb-Douglas)}$$

Where: $Y =$ GDP; while $\alpha + \beta = 1$ and $\alpha = \frac{2}{3}$ $\qquad C = (X_L)^\beta$ the low constant energy of labor

   $X =$ exergy (net useful energy) $\qquad K =$ capital $\qquad L =$ labor

   Note: Cobb-Douglas' Total factor productivity $A = C \times (X_K)^\alpha$ $\qquad$ (Keen et al., 2019, p 44).

   The current subject of this critique is the most recent paper, "Lotka's wheel and the long arm of history: how does the distant
past determine today's global rate of energy consumption?" ($Lotka's\ Wheel$) (Garrett et al., 2022). This paper presents a set of
observations, propositions, and conjectures that are thought provoking. This thermodynamic view models a society as similar
to an organism that starts from a single cell, and as it grows to adulthood, represents the sum of all of the work required for it
to grow and maintain itself (Garrett, 2011, pp 438-443)[2]. It is from this idea the $W$ concept comes, founded in the idea that
current society is built of everything that came before it.
This critique makes use of high quality data sources to assemble replicate datasets designated $X_{Rep}$ where $X$ is any symbol,
to examine the propositions and conjectures of $Lotka's\ Wheel$. Similarly, $X_{LW}$ means a dataset from $Lotka's\ Wheel$, where
$X$ is any symbol. The replicate estimates go back to year 1 CE, albeit with interpolated gaps. These are an improvement on
the supplementary datasets supplied for $Lotka's\ Wheel$, and are available in the dataset linked just above references at the
end of this article. The supplement of this critique can be viewed as showing differences from the idealized dataset of $Lotka's$
$Wheel$.

---

[1] I worked some on applying this equation and think it is still in development, though useful for theoretical concept.

[2] The meaning of the symbol $w$ for work in Garrett, 2011 "*Are there basic physical constraints on future anthropogenic emissions of carbon dioxide?*" is
not the same as the $w$ of $Lotka's\ Wheel$ Garrett et al. (2022). Thermodynamic work is the physics definition. In this economics use, production which is
technically the results of work, is used as a surrogate. Production is measured in dollars as GDP.





Beyond what I have assembled herein, others have begun to quantitatively explore energy and economics of humans in deep time. Freeman, et al. laid foundations for pushing the limits of understanding energy consumption in human societies back 10,000 years, which presents the possibility to correlate it numerically with energy consumption in nearer time periods that are better characterised (Freeman et al., 2018). Smil discusses human energy/economy in the frame of 300,000 to 10,000 years

BCE (Smil, 2004). However those works are yet incomplete, so we must wait until further work is available for deep time data.

$Lotka's\ Wheel$ uses GDP as GDP for the globe, which is not unprecedented, but can be confusing. Here I use gross world product ($GWP$). $Y$ is the symbol used in equations of $Lotka'sWheel$ for $GWP$, and is commonly used generically in economics to represent production, and I use $Y$ in the same way. Both $Lotka's\ Wheel$ and this critique are cross domain analyses. Features like use of $\pi$ as a symbol for inflation will seem strange to scientists, but is the usual nomenclature for

economics.

## 1.1 Background

Figure 1 provides a foundation for understanding the two underlying issues of $Lotka's\ Wheel$. These are:

1. What is the optimal equation for representing the curve of $\frac{W}{E}$ between 1970 and 2019? Can it reasonably be a represented by a constant? And do we see empirical evidence that should be present if representing it by a constant were true? What

are the implications of an optimal representation that is not constant?

2. Does the $\frac{W}{E}$ equation for 1970 to 2019 hold true going back into the near and deep past?

From these foundations, whether the proposition in $Lotka's\ Wheel$, that an equation fitting $\frac{W}{E}$ is a law holding true going back in time can be determined. Whether $\frac{W}{E}$ will hold true for industrial society in the future is a conjecture.

## 1.2 The constant monetary ratio conjecture

The primary proposition contained in the $Lotka's\ Wheel$ equations is that there is an approximately constant ratio between historical cumulative production ($W$) (measured as GDP in USD) and energy consumption. This is also expressed as a claim of constant scaling of $\frac{W}{E}$ (Where: $E$ = exajoules yr$^{-1}$). This is proposed to be in force going back thousands of years, and that the "long arm of history" defines what $\frac{W}{E}$ can be. The thermodynamic logic of this is quite compelling and the relationship was first noticed in an effort to find constraints on the economic growth rate in order to use the constraint(s) to derive a limit

on CO$_2$ emissions for atmospheric science (Garrett, 2011).

$Lotka's\ Wheel$'s cumulative production function[3] (eq. 1) can be thought of as the numerator of an ultimate trailing average, in which the $n$ of years in this trailing average can be all of human history. This foundation equation (eq. 1) is the summation of $GWP$ over $n$ consecutive years, where $n$ can be very large, which is also expressed as an integral, although no equation is

---

[3]For ease of cross-refererencing, equations 1-4 share the same numbers in this critique as their designations in $Lotka's\ Wheel$.



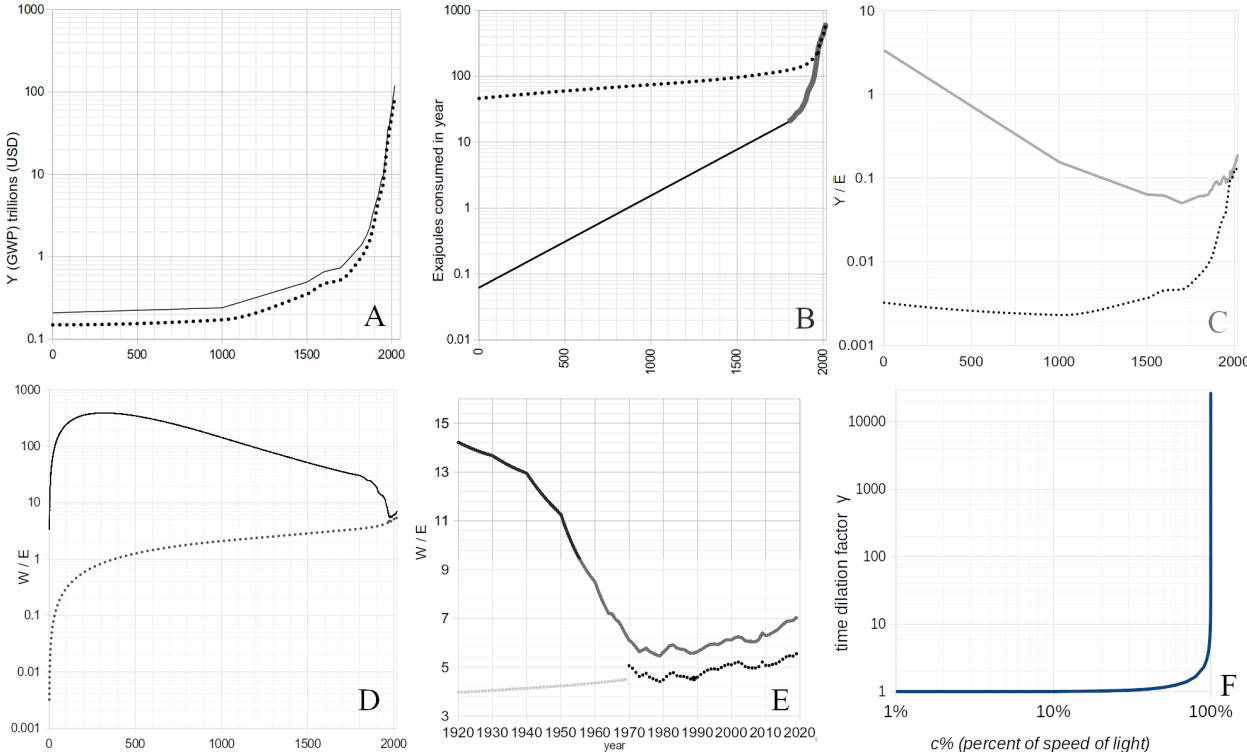

**Figure 1. Panels A-D** log scale, **panel E** linear scale. Here dotted curves are datasets from Lotka's Wheel supplement. Solid curves are replicates. **Panel A** shows $Y$ comparison. Dotted curve is $Y_{LW}$. Solid curve $Y_{Rep}$ is the interpolated composite. $Y_{Rep}$ is reasonably close, so this is not a primary reason for a large discrepancy. **Panel B**. Exajoules consumed by year. Solid heavy curve is $E_{Rep}$ from 1800 to 2019, which dataset is high confidence. From the 1800 point of roughly 20 exajoules, $E_{Rep}$ is shown as a thin interpolation to year 1. The dotted curve is $E_{LW}$, which appears to have been generated from $Y_{LW}$ to back-propagate based on the equations created for the 1970-2019 dataset. **Panel C**. This shows the effect on the $\frac{Y}{E}$ ratio, which indicates that $W$ (the cumulative sum of $Y$ where $Y = \sum_{i=0}^{n} GWP_i$) will also show a large difference between $W_{LW}$ and $W_Rep$, as shown in D. **Panel D**. Here we see that the solid $\frac{W_{Rep}}{E_{Rep}}$ curve is much different than the idealized dotted $\frac{W_{LW}}{E_{LW}}$ curve produced from eq. 1. **Panel E** is detail of panel D. Solid curve is $\frac{W_{Rep}}{E_{Rep}}$ curve. The dotted black curve is $\frac{W_{LW}}{E_{LW}}$ from 1970-2019. The dotted gray curve is the projected value of the $Lotka's Wheel$ supplement $\frac{W}{E}$ prior to 1970. **Panel F** Log log scale. Time dilation gamma ($\gamma$) as a function of velocity relative to speed of light. If the proposition of Sect. 2 that when $\frac{dE}{dt} = 0$, then $Y = 0$ is true, then this graph approaching the speed of light is an example of the asymptotic behaviour of inflation that $\frac{dE}{dt}$ approaching zero should display.

defined for $Y$ in the integral.

$$W(t) = \sum_{j=1}^{t} Y_j \Rightarrow W(t) = \int_{0}^{t} Y(t)dt$$

Where: $W =$ historical sum of all $Y$      $Y =$ GWP      $t =$ time in years      (1)




Built upon equation 1 is a primary observation that there is an approximately constant ratio ($w$) between $W$ and $E$ per equation 2. Here, I slightly change equation 2 to $\overline{w}$ for clarity that it is the mean average. Equation 2 is then used to relate $\overline{w}$ to energy as $W = \overline{w} \cdot E$ (eq. 2.1).


$$\overline{w} \approx \frac{W}{E} = 5.50 \pm 0.21$$

Where Units: $E =$ exajoules yr$^{-1}$        $W =$ currency ( USD )        $\overline{w} = \dfrac{\text{currency ( USD )}}{\text{exajoule}}$        (2)

Solve equation 2 for $W$ (eq. 2.1).

$W \approx \overline{w} \cdot E$        Where: $\overline{w} =$ mean average        **(2.1)**

Based on inspection of Garrett et al. (2022, p 1023, fig. 2C) this trailing average $\overline{w}$ observation appears reasonable enough on
its face. However, by inspection of the replicate datasets of figure 1, it does not appear $\overline{w}$ is a constant. By inspection, curve
$Y$ (fig. 1-A) exceeds exponential growth. The $E$ curve (fig. 1-B) is interpolated by an exponential from 0-1800 CE, and then exceeds exponential growth. Additionally, $\overline{w}$, which is $\approx \frac{W}{E}$ (fig. 1-D & E) is seen by inspection to be an increasing function for both the $\frac{W_{LW}}{E_{LW}}$ and $\frac{W_{Rep}}{E_{Rep}}$ curves from 1970 forward. After 1970, the $\frac{W_{Rep}}{E_{Rep}}$ curve is slightly steeper than the $\frac{W_{LW}}{E_{LW}}$ curve.

### 1.3 The proposition that rate of change of energy drives inflation

The simplified $\overline{w}$ of equation 2.1 is then substituted in for $\frac{W}{E}$, as shown below, yielding equation 3.

$$Y = \frac{dW}{dt} \text{ per } W \text{ in eq. 1.} \qquad \overline{w} := \frac{W}{E} = \frac{dW}{dE} \Rightarrow \frac{W}{E} \cdot \frac{dE}{dt} = \frac{dW}{dE} \cdot \frac{dE}{dt} \Rightarrow \frac{W}{E} \cdot \frac{dE}{dt} = \frac{dW}{dt} = Y \Rightarrow Y = \overline{w} \cdot \frac{dE}{dt}$$

$$Y = \frac{dW}{dt} \Rightarrow Y \approx \overline{w}\frac{dE}{dt} \qquad\qquad (3)$$

Solve equation 3 for $w$ to estimate $\widehat{w}$.

$$\widehat{w} = \frac{Y}{dE/dt} = 5.9 \pm 2.2 \qquad \text{Where: } \widehat{w} \text{ is the estimate of } \overline{w} \qquad\qquad (4)$$


Based on equation 4, the proposition is presented that if $\frac{dE}{dt} = 0$, then:

**Quote 1.3**. "...real-world economic production disappears: that is, Y = 0. ...note however, that zero real, inflation-adjusted production does not forbid nonzero, positive nominal production. If there is a large difference between the nominal and real GDP, it appears in economic accounts as high values of the GDP deflator or as hyperinflation."
(Garrett et al., 2022, p.1024)

This proposition (quote 1.3) is presented as fact, and justifies several pages of discussion.





### 1.4 An apparent errata

**Quote 1.4**. "Cumulative production $W_i$ increased more slowly than $Y_i$ or $K_i$ by a factor of 2.7 over the 50-year period. This ratio is nearly identical to the factor of 2.8 increase found for $E_i$." (Garrett et al., 2022, p. 1022)

Proposed correction: "Cumulative production, $W_i$, increased 2.7X over the period 1970-2019. This is nearly identical to the 2.8X increase found for $E_i$. Whereas $Y_i$ with a 4.5X increase, grew at 1.7 times the $W_i$ increase over the 50 years. (This discussion of the 50-year ratio is referenced below.)

### 1.5 New datasets created for this analysis

The supplementary data for $Lotka's\ Wheel$ is quite complete, with an Excel file containing columns with worked examples.
The first tab is titled "Reconstruction," which after examination of certain data suggests that the name conveys that data prior to 1970 was interpolated or projected backward using the equations above, presumably due to lack of data. For the purposes of this critique, there are four replicated datasets: A. a best effort for gross world product ($GWP$) back to the year 1 CE which is the $Y_{Rep}$ dataset for this study; B. a best effort for exajoules consumption by year going back to year 1, which is $E_{Rep}$. C. Two new $W$ datasets $W_{LW}$ and $W_{Rep}$ are the result of using equation 1 to perform summation.

The provided supplement spreadsheet in the link at the end contains more columns created in the process of this critique.

#### 1.5.1 $GWP$ interpolated composite dataset—$Y_{Rep}$

The primary dataset for $Y_{Rep}$ was obtained from Our World in Data (2023b, (OWID)). This dataset has complete sequential data starting 1990. From the year 1 to 1960, intervals of 1000, 500, and 100 years were interpolated using growth function eq. 5.

$$N_i = N_0 \cdot a^x \quad \Rightarrow \quad a = e^{(ln(\frac{\frac{N_y}{N_0}}{x}))}$$

Where: $i = 0 \leq x \leq y$    $N_y$ is the end interpolation range value    $N_0$ is the start interpolation range value.    (5)

The years from 1820 to 1960, have linear interpolations over the 10 to 30 year intervals.

For the years 1960 to 1990, the FRED dataset NYGDPMKTPCDWLD (FRED, St. Louis Federal Reserve Bank, 2024) was used to estimate GWP. For this period, the OWID dataset had 10 year intervals. Each of the 10 year interval years for which
an OWID value existed were scaled against FRED values to generate a scaling multiplier. These multipliers were interpolated linearly over the 10 year interval and used to generate $GWP$ values over the intervals of the OWID data. See figure 1-A.

The OWID 2024 $Y_{Rep}$ dataset averages 1.44 times the $Lotka's\ Wheel\ GWP$ ($Y_{LW}$), with $\sigma = 0.06$. (Fig 1-A) This scaling discrepancy should not have anything read into it, as the $Y_{LW}$ dataset is not an error. The discrepancy is due to a change in how OWID $GWP$ and its data sources are calculated versus other datasets, and the difference is consistent enough.



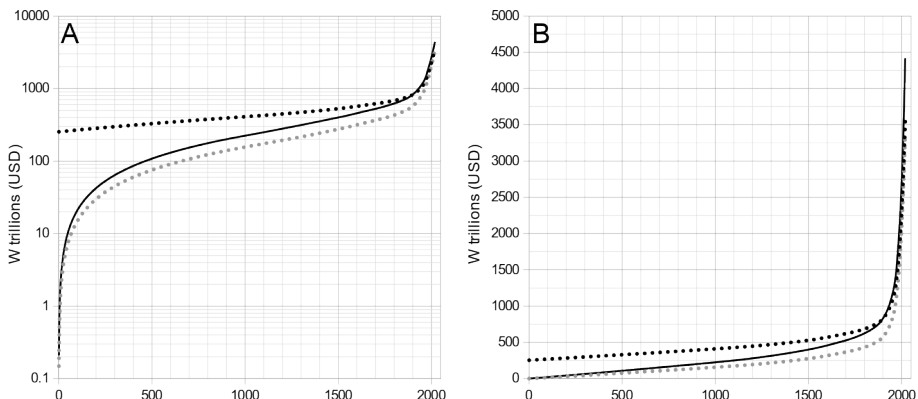

**Figure 2.** $W$ curve. A. Log scale. B. Linear scale. The upper black dotted curve is the $Lotka's\ Wheel$ supplement W dataset. The lower gray dotted curve is $W_{LW}$ for 1-2019 CE created per equation 1. Solid curve is $W_{Rep}$ for 1-2019 CE, also created per equation 1. What is visible here is that the algorithm of equation 1 when applied to generate $W$ from $Y_{LW}$ and the $Y_{Rep}$, data produces the expected curves that begins with the year 1 CE value. The provided $Lotka's\ Wheel$ supplement column labelled as $W$ does not. I suspect that the supplement W dataset of the upper curve is implicitly assuming a long projection back 10-20,000 years.

### 1.5.2 $W$ dataset in column provided by $Lotka's\ Wheel$ appears to be a summation of a much longer term dataset.

Two new $W$ datasets were made using the algorithm of equation 1. The first is a $W_{LW}$, the lower dotted gray curve in figure 2. The second is $W_{Rep}$ which is the summation of the $Y_{Rep}$ dataset, the solid curve of figure 2.

The $W$ dataset provided in the $Lotka's\ Wheel$ supplement starts at a value that is impossible given the supplied starting value of $Y_{LW}$ or $Y_{Rep}$ in year 1 (Fig. 2). (However, $W_{LW}$ and $W_{Rep}$ datasets have an artifact prior to $\approx 500$ CE, because a complete $Y$ dataset should start tens of thousands of years prior.) This indicates the supplied $W$ dataset is the projection of an equation 1 curve back into deep human historical time, and not produced from the $Lotka's\ Wheel\ Y_{LW}$ data provided. There are currently no published datasets/methods for determining $Y$ in deep time (prior to 1 BCE, and extending to 10,000-300,000 years into human past), although foundations are being laid (Freeman et al., 2018; Smil, 2004), and the next 5-10 years is likely to see major work.

Consequently, I believe that the $Lotka's\ Wheel$ supplied dataset was generated from a set of Y values obtained from a projection back into deep time intercepts, probably those mentioned in quote 1.5.2.

> **Quote 1.5.2.** "A least-squares fit to the logarithms of W and E yields the relationship W = 5.47E 1.00 . Calculated instead as a linear fit, the relevant expression is W = 5.67E - 66. Note the intercept of the fit, where E = 0, is equivalent to W = -66 USD trillion (2019)... " (Garrett et al., 2022, p. 1023)

### 1.5.3 Population dataset

For the year 1 CE, the $Lotka's\ Wheel$ dataset provides figures of 225.82 million people. The human population in year 1 CE from other sources is a range. OWID provides a figure of 232 million (Our World in Data, 2023a), and census consensus





provides a range of 140-400 million, with a mean and median of 270 million (United States Census Bureau, 2022). The $Lotka's$ $Wheel$ figures for population are accepted as reasonable.

### 1.5.4 Exajoules consumption composite dataset—$E_{Rep}$

The base replicate dataset for $E_{Rep}$ was obtained from Our World in Data (OWID) (Ritchie and Rosado, 2020). OWID based this dataset on a composite of of sources. The OWID figures are given in terawatt hours (TWh) separately by energy source. A column was inserted for sum of sources, and a column to convert TWh to exajoules (EJ) using a conversion of 0.0036 EJ per TWh. The OWID dataset has 10 year increments from 1800 to 1960, a 5 year gap from 1960-1965, and then is yearly. From 1800-1965, data was interpolated linearly in the gaps.

The $Lotka's$ $Wheel$ $E$ dataset ($E_{LW}$) in 1 CE is 46.17 exajoules, which is 12.8 petawatt-hours—the global energy consumption attained in 1902, which is not a reasonable value (fig. 1-B). Thus it was necessary to estimate this value.

For the year 1 CE, the number of watts available per 10 people was estimated from Smil (Smil, 2004, p. 552, figure 2) at 210 watts of continuous power, based on one horse-equivalent for every 10 people. Human labor was excluded as it is for all other parts of this thread of analyses. The work hours per day of this energy was set at 12 hours, for 252 watt-hours/person-day. A work year of 6 days per week, subtracts 52 days per year, and another 8 holidays per year yields 305.25 work days per year, for a total of 76,923 watt-hours per person year. Assuming a population of 225 million results in an estimate of 6.23E16 joules in the year 1, or 0.0623 EJ.

This figure of 0.0623 EJ was used as $N_0$ in equation 5 where $N_{1800}$ (the $N_y$) was known to be 20.3508. Solving for $a$ yields 1.00322299, which is an average annual growth rate of ≈0.322%. See comparative results in figure 1-B.

For discussions below, EJ means exajoule and $E$ is an exajoule dataset.

## 2 Examining evidence for the energy rate proposition that if $\frac{dE}{dt} = 0$, then $Y = 0$.

If this proposition that when $\frac{dE}{dt} = 0$, then $Y = 0$ is true (see eq's. 3 & 4), then there should be evidence of severe inflation caused by $\frac{dE}{dt}$ being near or equal to zero in empirical data. I will assume in this analysis that for inflation to go from positive to negative, it must pass through zero.

If this proposition is true, then there should be hyper-exponential increase of the denominator as $\frac{dE}{dt}$ approaches zero. An example of a physics quantity that exhibits hyper-exponential behavior follows (Sect. 2.1).

If the proposition is true, there should also be a visible relationship between CPI inflation and YoY $\frac{dE}{dt}$ in empirical data that suggests corroboration.

### 2.1 Expected behaviour of inflation as it approaches and crosses the zero boundary, if $\frac{dE}{dt} = 0$, then $Y = 0$

$Y$ is production, which is synonymous here with $GWP$ expressed in real (2019) dollars. $Lotka's$ $Wheel$'s proposition says that a system in balance having a growth rate, $g = 1.0$, and $\frac{dE}{dt} = 0$ should experience inflation ($\pi$) such that a zero real $GWP$ is obtained.





Inflation acts as a divisor of monetary value. So this proposition requires that a "nominal" GDP in USD, $x$, goes to zero
by virtue of an inflation numerator, $\pi$. Thus, for any non-zero $x$, $\lim(\frac{x}{\pi} \to 0)$ coincides with the $\lim(\pi \to \infty)$. The numerator
has a limit at infinity, but $x$ never becomes zero, only approaches it. This should mean that the inflator should be extremely
large relative to the denominator. It also means that the inflator should show hyper-exponential growth as it asymptotically
approaches zero.

It is hard to find legitimate use examples of quantities going to infinity, but time dilation of special relativity is one. Time
dilation prevents an object with a mass at rest greater than zero being able to quite reach the speed of light, because time
dilation's gamma ($\gamma$) approaches infinity (Libretexts Open Physics, 2024).

$$\gamma = \frac{1}{\sqrt{1 - \frac{v^2}{c^2}}} \tag{6}$$

Just as the velocity of light limit explored in figure 1-F is never attainable for any object with a rest mass greater than zero
due to time dilation, the zero real $GWP$ of equation 4 should not be attainable. (i.e. the zero bound should be impossible to
cross.)

## 2.2  The growth equations do not show evidence of discontinuity at $\frac{dE}{dt} = 0$.

$Lotka's\ Wheel$ [p. 1022] uses a 50 year ratio of cumulative production increase, as discussed in quote 1.4. Here is examined
how such a cumulative production function .generally behaves using the class of equations defined by equation 7.

$$Y = g^t \Rightarrow \frac{dY}{dt} = g^t \cdot ln(g)$$

Where: $g$ = growth rate (1+r)          r  =  some variable rate < 1

t  =  year starting from 1

$$\tag{7}$$

The flatline curve for the equation $Y(t) = 1.0^t$ corresponds to the flatline Y of quote 1 in section 1.3. For this flatline curve,
no growth is occurring in a stable economy.

$$\frac{dY}{dt} \text{ where } Y(t) = 1.0^t \Rightarrow 1.0^t \cdot ln(1.0) = 0$$

Note:  In the real world $E$ is a limiting variable input to $Y$. (Keen et al., 2019, p 44)

$$\text{Therefore, if } \frac{dE}{dt} = 0 \text{ then } \frac{dY}{dt} \approx 0. \text{ Ceteris paribus, if } \frac{dY}{dt} = 0 \text{ then } \frac{dE}{dt} \approx 0. \tag{8}$$

This flatline equation, eq. 8, is a special case for growth function eq. 7 that reduces to zero in a system where g is variable.
This equation contradicts the claim that $Y$ necessarily reduces to zero when $\frac{dE}{dt} = 0$. In this instance, $Y = 1$, and this is what
one would expect for a flatlined zero growth economy.





Note that according to figure 1-C, the trend since 1700 should mean that for a steady state $Y$ in the modern world, $\frac{dE}{dt}$ should be declining. This is another argument against this secondary hypothesis that when $\frac{dE}{dt} = 0$, inflation sends real GWP to zero. Because $Y$ should show an upward trend, when $\frac{dE}{dt} = 0$.

## 2.3  Empirical test of year on year (YoY) $\frac{dE}{dt}$

Let us examine a comparison (see: fig. 3) of YoY inflation figures against YoY $\frac{dE}{dt}$ computed as $\frac{E_i}{E_{i-1}}$. I will assume that if the proposition that $\frac{dE}{dt} = 0$, then $Y = 0$ is true, then it should show some signs of significantly correlated behavior near and below the zero boundary. Here, significantly correlated means that inflation spikes dramatically suggesting hyper-inflation approaching the zero boundary, and this behavior is consistent.

The first interesting change in YoY $\frac{dE}{dt}$ in figure 3, is for years 1974-75. These two years, $\frac{dE}{dt}$ dips to 1.5 and 1.8, and
inflation rises from 6.2% in 1973 to 11% in 1974, then drops to 9.1% in 1975. Next, a dip below zero $\frac{dE}{dt}$ to -2.3, -0.85, and -0.99 in 1980-82 accompanies inflation of 13.5%, 10.3%, and 6.1%, respectively which is on the high side, but certainly not hyper-inflationary. 2009 has the lowest $\frac{dE}{dt}$ of the dataset at -8, and inflation is -0.36%. These data do not appear to support the proposition.

$Lotka's\ Wheel$ has awareness of the difficulty of resolving "short-term" behaviours using the model, per quote 2.3, because
$W$ results from a smoothed function. However, in this section (see: fig. 3), we are not actually referencing $W$, but instead, we reference $Y$ in context of eq. 4. The claim that $Y$ goes to zero when $\frac{dE}{dt}$ is zero, is presented in context of the 50-year $Lotka's$ $Wheel$ study period composed of the discrete data of $E$ and $Y$, which means that those discrete datapoints should conform to the claim.

**Quote 2.3**. "Equation (3) assumes only that $w$ is a constant, a result that can be readily refuted, or supported
as done here, with decades of data from multiple sources. The approach does nonetheless have some important limitations, notably an inability to resolve short-term, fine-scale behaviors. The evolution of cumulative inflation-adjusted world economic production W is highly smoothed because it is a summation, or integration, over history and the global economy. Even given a strong multi-decadal relationship of E to W, year-to-year variability in E, such as during recessions or pandemics, cannot be easily related to yearly economic production, especially on
national or sectoral scales much smaller than the world as a whole." (Garrett et al., 2022, p. 1022)

One can argue regarding figure 3 that a half-century is not a large enough dataset. However, $Lotka's\ Wheel$ is concentrated on this half-century, so if the dataset isn't large enough that also may undermine the primary thesis of $Lotka's\ Wheel$. One can also argue that $\frac{dE}{dt}$ being zero, near, or below zero is not as prolonged as it would be in a system in balance. However, there is a pair, 1974-75, and a triplet 1980-82, that are close to, and below zero. Those are, therefore, extended periods of 2 and 3
235   years out of the 50. One can argue that $\frac{dE}{dt}$ has units of exajoules and so even small values of exajoules are still very large, as an exajoule is $10^{18}$ joules. However, because $\frac{dE}{dt}$ is negative for 1980-1982, and 2009, $\frac{dE}{dt}$ should have passed through zero at least 4 times. This should have exerted effects if the claim were true, which needs to be explained.





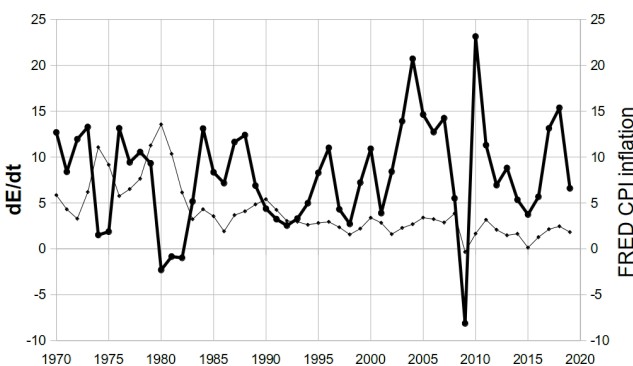

**Figure 3.** YoY $\frac{dE}{dt}$ compared to CPI. YoY $\frac{dE}{dt}$ (heavy curve) change against FRED CPI inflation (thin curve) (FRED, St. Louis Federal Reserve Bank, 2024). Here we see that there is no consistent or significant relationship between $\frac{dE}{dt}$ and inflation. There is no suggestion of hyper-exponential increase of inflation near zero. When $\frac{dE}{dt}$ crosses zero inflation might rise, or it might fall. In 2009, inflation went to zero when $\frac{dE}{dt}$ dropped precipitously below zero.

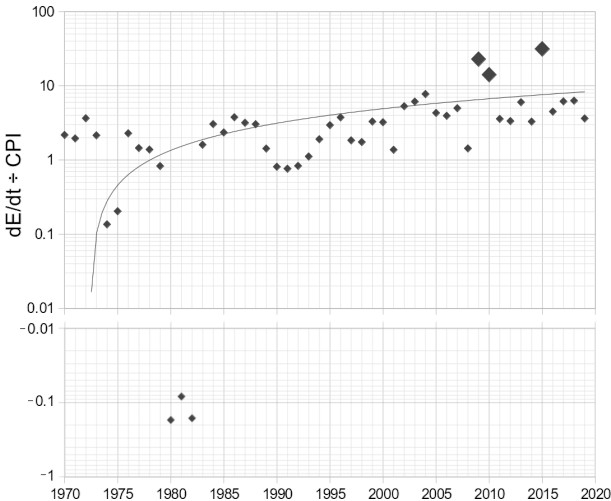

**Figure 4.** YoY $\frac{\frac{dE}{dt}}{CPI}$. Log scale. Note 3 outlier points with an absolute value greater than 10 (large diamonds), and outlier negative values.

Recall that per the discussion in Sect. 2.1, regarding the speed of light example, $\frac{dE}{dt}$ under $Y$ in eq. 4 should not actually reach nor pass through zero—and yet it does.

**2.4 There is an empirical relationship between CPI and dE/dt, but it does not justify the proposition presented.**

Figure 4 shows what appears to be a relationship of $\frac{\frac{dE}{dt}}{CPI}$ that has an increasing linear trend, barring extraordinary events like the global financial crisis of 2008. Excluding all data with an absolute value less than 10 leaves three outlier points, 2009-10, and 2015.





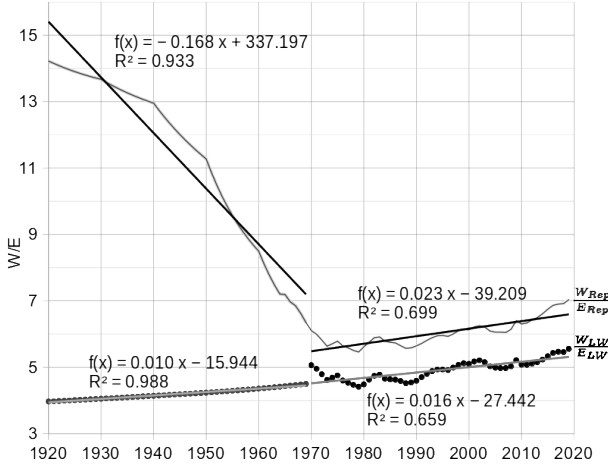

**Figure 5.** $\frac{W}{E}$ detail 1920-2019. Linear scale. Solid grey curve is the $\frac{W_{Rep}}{E_{Rep}}$ dataset. Dotted curve is the points of the $\frac{W_{LW}}{E_{LW}}$ dataset. Note the artefact where the projected data to 1969 comes in below the real data of 1970. From 1920-1969, the $\frac{W_{Rep}}{E_{Rep}}$ slope is roughly -16.8% per year, while the $\frac{W_{LW}}{E_{LW}}$ has a slope of $\approx +1\%$. From 1970-2019, $\frac{W_{Rep}}{E_{Rep}}$ slope is $\approx +2.3\%$ while the slope of the $\frac{W_{LW}}{E_{LW}}$ is $\approx +1.6\%$. For comparison, for the period from 2010-2019 the yearly growth rate of the US economy held to a range between 1.55% and 2.95% (World Bank, 2024).

$Lotka's\ Wheel$'s interpretation of equation 2 appears to say that there is an ongoing maintenance cost that must be surpassed in order for inflation adjusted GDP to be greater than zero. I do not see any support for this proposition that holds up. I view this positively.

## 3 Pre-1970 $E_{LW}$ figures are presumed back-calculated, with a 1970 artefact.

In the comparison of the whole exajoule dataset (fig. 1-B), the 1800 CE $E_{Rep}$ value of 20.35 EJ, which is a strongly supported point in the OWID composite, is just 16.3% of the $E_{LW}$ value of 124.6 EJ in the same year. In fact, this 1800 CE value of 20.35 EJ is approximately half the $E_{LW}$ value in the year 1 CE. Similarly to the way that figure 2's $W$ curve can be explained by back-projecting based on data from $\approx 1970-2019$, the exajoule dataset was probably produced the same way. Back-projection is also supported by the $E$ artefact in 1970 visible as an inverse artefact in the $\frac{W}{E}$ datasets comparison of figure 5 as there is a break-transition from the variation characteristic of real data, to a smooth curve.

## 4 The $\frac{W_{LW}}{E_{LW}}$ and $\frac{W_{Rep}}{E_{Rep}}$ figures are remarkably different prior to 1970

Figure 1-D shows great divergence of $\frac{W_{Rep}}{E_{Rep}}$ compared to $\frac{W_{LW}}{E_{LW}}$ values of $\frac{W}{E}$. To explain this, figure 6 shows that in the progress of the industrial revolution, real $Y$ increased faster than energy consumption did after 1700. Prior to 1700, the amount of GDP per unit energy $\frac{Y}{E}$ fell steadily from year 1 to $\approx 1699$. The fluctuations from $\approx 1850$ forward suggest that more granular data





for previous years could show similar results. Providing $\frac{Y}{E}$ in figure 6 highlights greater than exponential improvement trend in utilization of energy since the industrial revolution began.

## 5 Discussion

While flawed, $Lotka's\ Wheel$ is an early attempt to work on some key unsolved matters in economics. These ideas are on the leading edge of economics, and there is a need in economics and finance for numerical, physics and energy grounded approaches such as this thermodynamics founded work.

One of the more intriguing secondary ideas of $Lotke's\ Wheel$ from an archaeological viewpoint is the deep time intercepts, which work out to be 30 kya (Garrett et al., 2022, p. 1023). The deeper and more accurately economic and energy consumption datasets can be defined from work such as Freeman et al., perhaps combined with geological holocene CO2 records, the more likely it is that back-projection into the past could be improved. However, before those are used, these equations need to be better understood.

### 5.1 Criticism

The reduction of eq. 2 to eq. 2.1 and then substituting eq. 2.1 into eq. 3, is a key error. It is incorrect to assume that this eq. 2.1 version of $W = \overline{w} \times E$ is generally meaningful for $Y$ in eq. 3. This is only usable as a "quick and dirty" approximation. This error leads directly to the claim of Sect. 2 that $Y = 0$ when $\frac{dE}{dt} = 0$, which claim is not supported by evidence. This error highlights that valuable theoretical approaches need empirical confirmation to fully work out what is going on. This appears to be overconfidence based on theory that is quite compelling. Reading (Garrett, 2011, pp 438-443), particularly p 442, helps clarify the strength of the theory that led to this. One can see why this constant could be seized upon as, "Aha!" But real economic data is more complex.

Building replicate long-term datasets for $E$ and $Y$ shows that these datasets are not supportive of certain specific claims of $Lotka's\ Wheel$. These datasets indicate that, rather than the "decades" time scale of Quote 2.3, time scales of centuries and if possible, millennia, should be the target for an effort of this kind. Thus it is my hope that this critique can be received positively as building on what $Lotka's\ Wheel$ began. Highlighting the discrepancies should be seen as a mystery to solve, rather than disproving the underlying theory.

I do not think that $\frac{W}{E}$ from 1970 to 2019 shows the long arm of history—it shows something else that may, perhaps, be expressed as a limit of history. Per figure 5, a superficial reading would suggest $\frac{W}{E}$ should continue to drop at -17% per annum or faster. A deep time dataset should generally resemble figure 1-D, but covering as much as 300,000 years. Thus, if one presumes that the basic thermodynamic argument is sound, then one must search for the factors that explain the shape of 1-D.

### 5.2 Strong elements of $Lotka's\ Wheel$

The most intriguing question is, how should we interpret that $\frac{W}{E}$ dropped from ≈300 CE to 1970, then stops like water hitting a riverbed? (See: fig. 5)




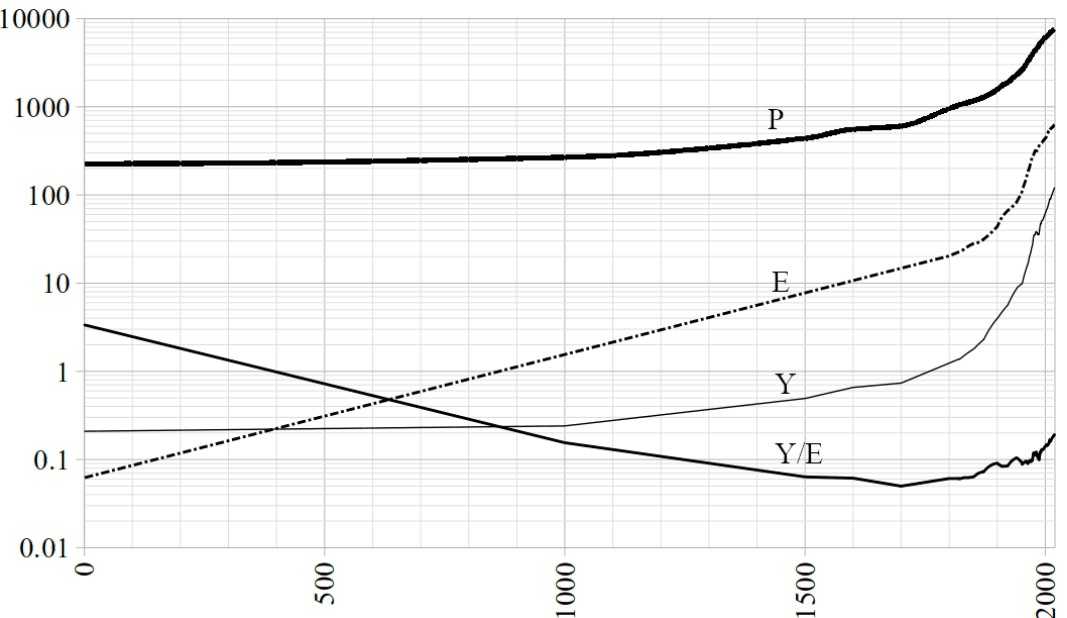

**Figure 6.** Y axis Log scale of replicates data. The behaviour of $\frac{Y}{E}$ suggests that the underlying thesis of $Lotka's\ Wheel$ broadly may be correct. Here is visible that from 1 CE to ≈ 1700 CE $E$ increases considerably faster than $Y$. $Y$ only provides estimated GWP for each year, which stays fairly flat, and then slow growth to 1700, despite rising energy expenditure, which suggests quite poor energy utilization. This is most visible in the $\frac{Y}{E}$ curve which shows exponential decline in GDP per unit energy. A caveat is that both $Y$ and $E$ are interpolated estimates and better historical estimates will probably show fluctuations not visible here, though the trends should remain. Note that the industrial revolution is roughly visible at 1850, but the turning point appears around 1700, coincident with increasing population. Solid medium width curve $Y/E = \frac{Y_{Rep}}{E_{Rep}}$ in trillions USD. Solid narrow curve $Y = Y_{Rep}$ in trillions USD. Dash-dot curve E = $E_{Rep}$ in trillions USD. Heavy curve P = population in millions.

In the most superficial sense, it means that after 1970 $GDP$ growth finally began to outpace growth in $E$ such that this

registered in the $W$ summation of $Y$ (fig. 6 & 5). However, this does not answer the question of whether our modern world must keep on an energy to GDP trajectory like this or pay a grievous price. It makes me uneasy that the relationship to entropy (decay of capital, $K$) may be visible here, in a continuous increase in $E$, although $Y$ is increasing relative to $E$. My concern is that too much or $Y$ could be churn.

Perhaps $Lotka's\ Wheel$ is on the right track that it has primary meaning relative to capital ($K$) maintenance. Keep in mind

that fig. 6 is a log scale, and behaviour since 1700 looks hyper-exponential. $Lotka's\ Wheel$ touches on, but does not delve into this. (viz. Missing decay rate for eq. 1.)

Figure 6 shows that primitive technology prior to 1700 was progressively more inefficient the more energy was available. The same figure also makes clear that since ≈1700 CE, $\frac{Y}{E}$ has an increasing growth trend that by inspection appears exponential until 1850, when an apparent hyper-exponential trend begins. This 1700 inflection point is interesting, perhaps backing up

the true start of the industrial revolution to Newton's time and the impact that calculus and physics had, applied to things as





apparently simple as plow design. The appearance of calculus, and major improvements in physics could be an example of educational, cultural and intellectual capital making a critical change.

Buckminster Fuller noticed this $\frac{Y}{E}$ trend of "more with less" in the 1970's, calling out economists for their tendency to a static view of goods and services based on education in STEM, and for the malthusian doctrine that necessarily follows (Fuller,
1981, p. 133, 213), (Fuller, 1975, p. 23-24, 26-27).

Net of inflation, GDP is increasing faster than energy use, which is a very positive observation for society. $Lotka's\ Wheel$ mentions this, although not as positively as I view it.

### 5.3   Observations on capital ($K$) and decay (entropy)

Some of the faster increase in $Y$ over the increase in $E$ may be a kind of fiction, because of capital ($K$) churn in the form
of value engineering and planned as well as opportunistic obsolescence. However, one could also argue that a society that correctly forecasts the social need for $K$ could be more efficient. To what extent this is true should come out in a careful accounting of $K$, and represents some of the entropy of the economy.

There needs to be agreement regarding what, exactly, $K$ includes—for instance does it include commodities from napkins to plastic forks, or is $K$ just the means of production? There is also the question of what the services economy means relative
to $K$. Additionally, this $K$ question gets into the realm of capital as defined by culture (global culture as well as national) and subcultures (i.e.. corporations, ethnic groups), education, institutional knowledge, entrepreneurial spirit, social pathways of contribution to the economy, social capital (one representation of which is the corruption perception index (Lambsdorff, 1999) ) and other "soft capital". All of these need to be accounted for, and estimated in deep time, which requires that a common definition for capital specific to the thermodynamic approach ($K_{Thermo}$) needs to be forged. This definition needs to
be something that can be grasped and estimated by archaeologists and categorized for those that work on this. I do not have answers for these knotty questions.

I think the general thesis presented in section 4 of $Lotka's\ Wheel$ relative to $K$ is worth pursuing. I did not pursue any work on $K$, which is less tractable than $GWP$ and energy consumption. When considering $K$, it is intuitively obvious that if physical capital, for instance roads, harbors, airports, spaceports, trains, dwellings, water pipelines and energy installations,
etceteras last longer, this can be higher overall energy efficiency than our current methods. Currently, many civil engineering projects, and certainly homes, are engineered for an assumed life span. For such civil engineering projects, value-engineering is not necessarily the same as the planned obsolescence that many manufactured products have engineered into them, even though it effectively generates the same net result of requiring more "churn" of $K$.

To planned obsolescence we should add opportunistic obsolescence. Apple, for instance, was caught deliberately draining
power and slowing down older phones, with the apparent intention to to sell new iPhones (Cooper, 2024). This product churn motive of our economic system is problematic. It signals mature industry that has few or no more innovations to offer that are valuable, or one with a degree of monopoly power that has decided to play rentier games with their products (Duenhaupt, 2012). Every industry tends to get to this point, even if the dominant corporations can be overthrown later for something cheaper, more efficient, or presenting new capability (Schumpeter, 1943, pp 81-86).





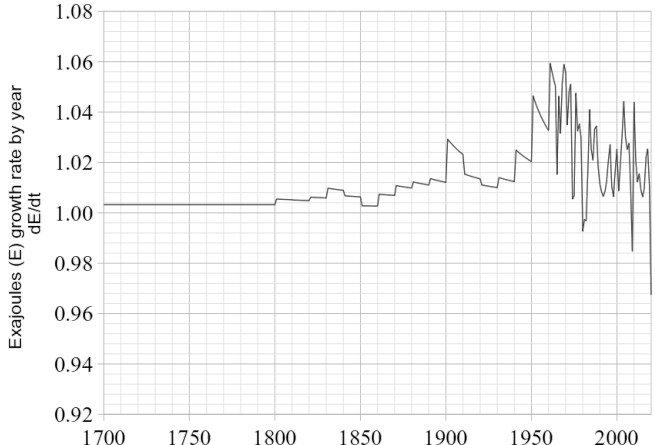

**Figure 7.** Growth rate of $E_{Rep}$, $\frac{dE}{dt}$. 1970 is the peak of the increasing $\frac{dE}{dt}$. After 1970, the trend is clearly downward for 50 years. What is the significance of this relative to the effects on $\frac{W}{E}$ and $\frac{Y}{E}$? Is it a driver of the graph of fig. 5 post 1970?

There are certainly challenges to increasing life span of physical capital dramatically. Our economic system is built on churning consumer products, and this improves economic activity, ignoring that churn shortens the lifespan of $K$, thus essentially increasing the rate of entropic breakdown. One is reminded of Keynes, "digging holes and filling them up" as the epitome of what could be called synthetic economic activity—churning while creating little value. We also still have the problems of rentier economics that can predominate when physical capital is stable over generations, and the societal stagnation that this

generates. Even though real interest rates show an 800 year decline (Schmelzing, 2020), the rentier problem Keynes decried (Keynes, 1936) seems to have increased (Christophers, 2022).

## 6  Concluding thoughts

After contemplating these curves, I will offer some observations. First, that in the real world individual civilizations rise and fall, hence $GWP$ over millennia in deep time is the collective of at least hundreds of civilizations. The Garrett thermodynamic

model should, like the growth of an individual person, also account for the fall of civilizations (Bardi, 2017). The Garrett model does seem to include senescence and ending, although human civilizations could theoretically grow and last an extremely long time.

It seems likely that while individual civilizations may conform to Garret's model, they may do so in different ways. For instance, at the same rough level of $E$, the industriousness, skills, literacy, education, and record keeping keeping can vary a

great deal.

Related to this multi-civilization model, I note that most analyses seem to leave out the $E$ of human labor and its exergy, as well as the earliest $E$. The energy of human labor involved in production was probably greater than or equal to that of horses, camels or other draft animals in most early societies. But the effectiveness of human labor at production ($Y$) doubtless varied



between societies. Even today, intensive manual gardening is 3 to 20 times the productivity per acre of modern mechanized
agriculture (Hervé-Gruyer and Hervé-Gruyer, 2016; Morel et al., 2017).

The earliest form of $E$ was the use of fire. Built on that, for instance, one of the earliest inventions was the atlatl, which allows a modern man to throw a fire-hardened wooden spear through a car door (or 300-50 kya, a mammoth hide). A society with such an invention has an exergy advantage. It seems possible that as the non-human $E$ fraction rose or replaced human labor, $Y$ could fall relative to $E$, if for no other reason than the removal of most of human $E$, which could be named $E_h$. This
might explain how $\frac{Y}{E}$ could decline for 1700 years as energy consumption increased exponentially (fig 6).

Second, perhaps this turn in 1970 (fig. 5) might signal the transition from a set of siloed less efficient civilizations, to a de facto single market global civilization. Proving whether such a relationship may exist is non-trivial and beyond the scope of this critique.

Third, it may be significant that 1970 is the rough peak of growth in the rate of global energy growth (fig. 7). 1973 was the
the debut of the OPEC oil cartel, which ushered in a period of forced higher energy efficiency on the global economy. This was particularly true of the United States, which was the largest economy in the world in 1970, larger than the next economy, the USSR, by 2.5 times. That the globe has not trended upward in its rate of energy growth since then may also be suggestive of a slowdown preceding the true Hubbert's peak (Tverberg, 2021a, b).

I had hoped that the $Lotka's\ Wheel$ paper would present some method(s) or correlation(s) to support either a time span
(perhaps as some $f(x)$) for the summation of equation 1, or else a decay factor in the integral. This was mentioned, but not provided (Garrett et al., 2022, p. 1022). I lean toward the integral of equation 1 probably having the most meaning when applied to capital ($K$), however, since all of this is clearly discrete data, one can argue against using an integral function. This $W$ dataset seems to be a kind of stand-in for physical, social, cultural, and educational capital over historical time.

This 50 year ratio of $\frac{W}{E}$ and the associated thermodynamic concepts are important, intriguing, and should be further inves-
tigated. I think that the methods of $Lotka's\ wheel$ represent an important beginning. The theory presented in (Garrett, 2011, pp 438-443) is so compelling that I think it is probably true, and we need to figure out what factors are causing the large discrepancies visible in figures 1-C, D, E, and figure 6. I do not think that the falsified elements disprove the foundation concept of the relation.

Deep time $GWP$ and $E$ datasets are on the verge of appearing and should prove helpful. Examination of capital ($K$) and
entropy's effect on capital, as well as sound definitions of capital that can be used by archeologists and anthropologists allowing estimation of $K$ into deep time, is in order. Developing better deep time datasets for understanding energy and production is a daunting, but likely possible, task. Coordination with archaeologists and anthropologists working in the area is needed.

*Data availability.*  Pre-publication Figshare link. Must be updated with publication DOI after acceptance. https://figshare.com/s/9870ad630c0bbee2e709
Original data is available as link from Lotka's Wheel paper. A copy of this spreadsheet is in the fiirst sheet of the figshare link.





*Author contributions.* This article was authored by Brian P. Hanley.

*Competing interests.* The author declares no competing interests.



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
