# Peer review of "Critique of 'Lotka's wheel and the long arm of history': Best available historical data shows major differences pre-1970, raising new questions."

_EGUsphere, 2025_

## Referee Comment (RC1)

The article "*Critique of 'Lotka's wheel and the long arm of history': Best available historical data shows major differences pre-1970, raising new questions.*" terms the original paper by Garrett et al. (2022) "flawed" (hereafter GGK22). It misrepresents both the data and methods in GGK22 and presents alternatives that do not reflect the analysis techniques that were performed in GGK22, justify alternatives, perform an uncertainty analysis, or even seem plausible. The critique includes personal asides, the mathematics is impossible to understand, it is left unstated in the figures what units are used for quantities, and when units in the analyses are presented, they are used inconsistently for the same quantities. There are frequent uses of very large numbers of significant digits for imprecise quantities, up to nine for an exponential growth rate in one case. Otherwise, I discuss the main methodological concerns below.

A primary conclusion of GGK22 as stated in the abstract is that "In each of the 50 years following 1970 for which reliable data are available, 1 exajoule of world energy was required to sustain each 5.50±0.21 trillion year 2019 US dollars of a global wealth quantity defined as the cumulative inflation-adjusted economic production summed over all history." That is the ratio $W$ for cumulative inflation-adjusted production $Y$ (or $W(t) = \int_0^t Y(t)\,dt' = \sum_i Y_i$), to the rate of energy consumption $E$, is effectively observed to be a constant for a 50-year period since 1970, that is $w = W/E = \int_0^t Y(t)\,dt' = $ constant. As is argued in the GGK22 article, this constancy, were it to hold, it would imply that inertia, i.e. deep history, plays a guiding role in the evolution of future consumptive needs by civilization, one far higher than has generally presumed.

The result that $w = W/E$ is a constant was originally anticipated based on thermodynamic reasoning as first described in Garrett (2011) and it was also elaborated upon in GGK22. As with any hypothesis, it's validity rests on a statistical evaluation, as well as the accuracy of the statistics used. In GGK22, a half century period between 1970 and 2019 was used to evaluate $w$. That the statistic holds for any period before or after can really only be speculation, but for that period covering two-thirds of the total historical growth in civilization energy demands $w$ was observed to be nearly constant. It was speculated that this constancy is a fundamental feature of the global economy: energy consumption sustains flows along the civilization networks that have previously been built through prior economic production, allowing for fraying of those networks by way of the downward correction from the nominal to the real GDP, as is commonly ascribed to economic inflation.

However, as a time integral of inflation adjusted economic production, the accuracy of the calculation of $W$ – and hence $w = W/E$ – relies on a reconstruction of historical economic data. Any reexamination of the statistics and the methods is important because any reconstruction necessarily requires some subjective choices. It is also challenging. An independent reexamination of the methods used in GGK22 could shed light on better approaches. B. Hanley provides revised reanalysis datasets from which he makes arguments that $w$ is not in fact a constant, in any time period.

To address the Hanley critique, I compare the reanalysis dataset in GGK22 to that proposed as alternative of Hanley, what he terms a replicate dataset. Statistics for world primary energy consumption since 1970 are stated clearly enough by both sources. Specifically as stated in the Methods of GGK22 *Yearly statistics for world primary energy $E_i$ are available for both consumption and production from the Energy Information Administration (EIA) of the US Department of Energy (DOE) for the period 1980 through 2018 and for consumption from British Petroleum (BP) for the years 1965 through 2019 (DOE, 2020; BP, 2020). A yearly composite of $E_i$ in units of EJ $yr^{-1}$ for the years 1970 to 2019 is created from the average of the three datasets while using single*

*sources when only one is available. The difference between the values in the BP and EIA datasets is significant at 8.5 ± 1.5 %, but it is steady and small relative to the 180 % increase in energy consumption over the 50-year time period considered here.* In the Hanley critique, no argument is presented why the energy data used in GGK22 is problematic. The data set Hanley used was obtained from Our World in Data (Ritchie and Rosado, 2020) which refers to the Statistical Review of World Energy. No averaging is made of consumption and production as in GGK22 but this is a small consideration. Based on that dataset, consumption in 2019 was 588 ExaJoules compared to 595 ExaJoules in the GGK22 dataset. Because this difference is small, it is presumed that the two analyses are in sufficient alignment even if the GGK22 datasets is more comprehensive.

However, Hanley shows a reconstruction of energy demands in 1 CE that is *three orders of magnitude* smaller than that which may be inferred from GGK22 assuming *w* is a constant. Where such large discrepancies exist it should be possible to perform a simple test to determine if one result or the other is unreasonable. The Hanley critique presents the global consumption rate of energy in 1 CE as 0.0623 ExaJoules, which split among the estimated 225 million people in the world at that time would correspond to a civilization consumption rate averaging to about 9 Watts per person. Human metabolic needs are approximately 100 Watts. To feed those needs using modern agriculture, the efficiency is about 10% implying order 1000 Watts of consumption. Presumably ancient agriculture was less efficient, and then of course not all primary energy at that time or any time went into feeding people in a subsistence lifestyle. Militaries, buildings, roads, and sailing networks were also needed. Whatever the source of the calculation, the 9 Watts per person presented by Hanley as part of a superior dataset reconstruction is simply far to small to be plausible, to enable any form of civilization sustenance, and by two to three orders of magnitude.

The problems are more of a basic scientific nature when it comes to calculation of cumulative inflation-adjusted world economic production summed over all history $W$. The approach outlined in the Methods section of GGK22 is that *Economic production is tallied and averaged using World Bank (WB) and United Nations (UN) statistics for the years 1970 to 2019 (The World Bank, 2019; United Nations, 2010) and expressed here in units of trillions of market exchange rate, inflation-adjusted "real"-year 2019 dollars.* So there are two key points here, which are that the production statistics must be adjusted for inflation and expressed in Market Exchange Rate (MER) values. The reason to do so was described in greater detail in a prior article referenced in the methods by Garrett et al. (2020) and in the paper the critique references by Garrett (2011). In the Hanley et al. critique, no criticism is made of statistics used or the choice to focus on real, MER dollars so presumably that is not an aspect in contention.

However, Hanley mixes and matches datasets to reconstruct a long time series for economic production that is inconsistent in its choice of units for each year, and so cannot be used for the purpose of addition to calculate $W$. The economic production statistics they use since 1990 appear to have as their original source the Penn World Tables, although details are not given. If this is the case, the both key points of MER, inflation-adjusted dollars are satisfied.

For data between 1960 and 1990, the Hanley critique states use of data from the Federal Reserve (NYGDPMKTPCDWLD). These are expressed in MER dollars but they do *not* account for inflation to convert from current-year nominal to fixed-year real dollars. This is explicitly stated in the Federal Reserve dataset but not acknowledged by Hanley. The description in Hanley is difficult to follow but it appears a scaling multiplier is applied to the Penn World Table datasets to match the two economic production datasets post-1990, and that the Federal Reserve data set is used as the basis for a longer time series. Not surprisingly, by not adhering to the requirement of using real

MER dollars for all years the result is an average 44% difference between the GGK22 and Hanley datasets. The difference is washed away by Hanley with "*The discrepancy is due to a change in how OWID GWP and its data sources are calculated versus other datasets, and the difference is consistent enough.*" This is misleading. For the purpose of calculating $W$, and error will be cumulative, in this case introduced by not considering the growing devaluation of purchasing power through inflation. Most importantly, for the purpose of calculating $W$, it makes no sense to sum dollars from one year to the next where they are expressed in the different units of current dollars from different years. The error is like adding 1 pint to 1 quart and saying there are two quarts.

For the reconstruction of GDP for earlier years, Eq. 5 is difficult to understand and highly contrived given $e^{\ln x} = x$. The reconstruction refers to Our World in Data datasets for GDP statistics pre-1960, which in turn appear to use the same Maddison database used by GGK22. This database appears to have been adjusted for inflation, which is good. However, what is not considered is the second necessary adjustment to convert from purchasing power parity dollars used in the Maddison database to market exchange rate dollars. Specifically, as stated by GGK22: *The dataset is adjusted for inflation and to convert from currency expressed in purchasing power parity dollars to market exchange units using as a basis for adjustment the time period between 1970 and 1992 for which concurrent market exchange rate (MER) and purchasing power parity (PPP) statistics are available.* An argument might be made that PPP dollars are preferable to MER dollars. I don't believe this would be appropriate for physical reasons. Whatever the choice, there must be consistency when creating a continuous time series of economic production. The final dataset for $Y$ Hanley uses is a mix of real, inflation-adjusted, MER, and PPP data. Without *consistent* careful adjustment for inflation and to MER dollars across the full 2000 year time series, the numbers in each year cannot be added to create the central quantity $W$. Any summation must use numbers with the same units. Not all dollars are the same.

As a note, the value of $W$ obtained by GGK22 for year 1CE was not a summation of a much longer dataset as stated in Section 1.5.2 of the critique. The method was stated in the Methods section of GGK22 to be something quite different, an inference from the GDP at that time assuming that $W$ and population were growing equally fast. Note too the discussion of uncertainties that might accompany this assumption. *The value for cumulative production in 1 CE. W(1) is obtained by assuming that W was growing as fast as population at that time at rate and that $Y(1) = R_W W(1)$. Population data from 1 CE and 1 century before and after suggest that global population was 170 million and growing at 0.059 % yr$^{-1}$ (United States Census Bureau, 2018). While there are inevitable uncertainties in the reconstruction of W as with any other, the yearly values of W since 1970 that are emphasized here cover two-thirds of total growth, so the calculations are more strongly weighted by recent data that are presumably most accurate. Thus, calculation of W, most particularly the conclusion that w is nearly a constant, can be shown to be relatively insensitive to uncertainty in the older statistics (Garrett et al., 2020).*

Even the datasets provided in the supplement of GGK22 are misrepresented, although it is unclear how or why. Figure 1E pretends to reproduce the GGK22 data. However, it shows a distinct positive trend in the time series between 1970 and 2019 for $w = W/E$. This is false. Figure 2c for $w$ in GGK22 shows a different time series with no clear trend. The value presented by Hanley for $w$ in 1970 is about 5 (although no units are given). In the GGK22 dataset provided in the supplement it is 6.1 inflation-adjusted year-2019 MER USD per ExaJoule per year. The difference may seem small but it is significant where constancy is claimed.

The mathematics in section 2.2 makes no sense. How is economic production $Y$ equal to a

growth rate $g$? How can a dimensioned rate $r$ be added to a dimensionless number unity? It also appears to refer to calculus but not in any way that is familiar or that leads to results that are comprehensible.

While I completely applaud taking a closer look at the reconstruction in GGK22, the critique by Hanley is far too idiosyncratic in its analysis to be of a standard I believe suitable for publication in an EGU journal. As a suggested path forward, I suspect that a reconstruction of the timeline of civilization energy consumption in the distant past might be made using a combination of ice-core carbon dioxide concentrations, a carbon cycle model, and an assumption of a mean fuel makeup that does not minimize the widespread reliance on wind, water, and sunlight. People built where they could sail, navigate rivers, and grow crops because they offered forms of primary energy that could be efficiently accessed.

Tim Garrett

---

## Editor Comment (EC1)

**A response to the Hanley submission critique of the Tim Garrett et al (GGK22) Paper "Lotka's Wheel and the Long Arm of History**

**Richard Nolthenius – Earth Futures Institute at UC Santa Cruz, and Chair Astronomy Dept of Cabrillo College, CA.**

I've done an independent investigation into the validity of the GGK22 and prior work on what I've come to call the "Power/Wealth Relation" (P/W); i.e. the constancy of the ratio of the inflation corrected sum total of all human production, and the current consumption rate of primary power. I can confirm that the PW relation is indeed essentially a constant over the period since 1970 - when yearly global data become available in the literature. I won't repeat Garrett's own criticisms of Hanley's contentions here, but I too find Hanley's contentions just as puzzlingly unsupported and unsupportable. I don't have any disagreements with Garrett's points.  I'll say too, that I've never met Tim Garrett, and we have no professional or personal relationship. I'm intrigued by this work; that's my connection, and my interest is its connection to related work I do for the Earth Futures Institute at UCSC.

 My interest is whether the P/W relation can be expected to remain valid, and whether it may even be an inevitable feature of human civilization's evolution.  If so, it should be part of any integrated assessment modelling by economists in forecasting future outcomes of different realistic climate policy. I've done my own investigations and public lectures on this subject,  and the results are consolidated in Presentations, including this longest version **here.** I'll refer to it below.

1. **Adding the Shadow Economy:** The thermodynamic arguments that this relation is reasonable and should not be unexpected, should rightfully include not just the denominated wealth in monetized transactions, but also the "shadow economy" of barter etc. The most complete work I've found on the size of the shadow economy, as of a few years ago, is from Elgin and Oztunali (2012). They find that the shadow economy has been a slightly declining ~23% fraction of the visible economy since 1970 (see slide #291). This slightly declining fraction, when incorporated into the Wealth and into P/W relation, actually improves the constancy of that relation.

2. **The P/W Constancy over different time scales:** One might wonder whether 1970-> present is long enough. Garrett makes a strong case that there has been large changes in energy and economic factors during that period and yet the constancy is quite impressive. If the P/W relation is valid even for short time scales, it imposes constraints on how energy efficiency should behave during recessions.  I find that

data from the Federal Reserve supports the contention that even during shorter time scales of a year or two, that the P/W relation is obeyed better than the data suggests (which uses just the official figures for GDP). (see slide #295). There is a "Recession / GDP Bias" which over-estimates GDP during recessions and underestimates a bit in post-recession (slides #309-310). It is seen in the two largest economies in the world, and shows that energy efficiency actually stops improving and goes into reverse during recessions, and the economic motivations that this should be true are fairly easy to appreciate – energy efficiency investment is the least important thing to be funding when you're in economic contraction; far more demandingly immediate is preserving (against $2^{nd}$ Law of Thermodynamics decay) what has already been created, and then in developing new energy to support that preservation, and then if possible, expansion of energy access, and only after that spending is done, does investment in improving energy efficiency pencil out. Why? Because investment in improving energy efficiency only pays off in the future, which is discounted rather strongly by economic policy people advising corporations and government.  Historically, globally we have improved energy efficiency of GDP by 1.1% per year, but that only makes profit sense if we can achieve an ROI of 2.2% year growth in GDP. When GDP goes into recession, then energy efficiency, according to the P/W Relation, must reverse. A closer look at the data (slides #93-#97) shows that indeed it does (slides #309-311).

3.  **The PPP vs MER argument:** I'm in agreement with Garrett that MER is the proper method to calibrate between currencies. PPP is not. To use the "Big Mac Index" metaphor; that Big Mac has more capacity to improve labor production when in a country with stronger networks supporting that civilization, than in a poorer country. It's not just the income base that is the issue, it's the ability of that Big Mac to fuel production. Currency traders put money on the line, and are the more properly motivated actors to do the job correctly. Over very short terms, of course, traders' profit concerns can dominate, with resulting volatility, but averaged over a ~year, MER will be the more defensible measure of the calibrated value of a currency.  The data show that MER accounting shows slower economic growth among poorer countries than PPP accounting does, and this means that the energy efficiency improvements globally are not as impressive as PPP accounting would suggest (slide #307). This may be inconvenient to some. But more to the point, one can argue that the PPP vs. MER debate is not relevant for criticizing the P/W Relation. If the P/W Relation is obeyed only with MER accounting, so be it.   Further, if P/W isn't obeyed with PPP accounting, where instead it shows a slope, then this just might be a sign that it is PPP accounting that is flawed… if, as physicists do, we take observational patterns as trying to tell us of important truths we may not yet

appreciate, and which may contradict our ideology or assumptions, and so we should pay closer attention.

4. **Inflation measure:** Garrett uses the standard published dGDP (GDP deflator) inflation adjusted figures. But there is much debate about the way economists have chosen to re-define CPI and hence dGDP inflation with a change in method in the mid 1990's, and in a way which underestimates true inflation (slides #266-277) arguably for political reasons as pointed out by many. I've chosen to use the MIT's "Billion Prices Project" work to adjust dGDP, as a fairer measure (slide #283). When I do so, the P/W Relation is not only still supported, it is a bit better supported in its constancy.

5. **Civilization Obeys Thermodynamic Laws**: That human civilization systems obey thermodynamic laws is even more strikingly confirmed by the work of Victor Yakovenko (see slides #15-24 in this Presentation, with links to his publications) with the demonstration of the Boltzmann-Gibbs distribution of incomes, the close parallel of "income" with thermodynamic "temperature" and implications for entropy evolution and the Gini Coefficient in civilization's economic systems.

6. **Thermodynamics and Human Agency:** Finally, economists have, over the years, shown an antipathy with any connections with thermodynamics. Given Neoclassical ideology, a plausible explanation for this attitude is because it casts doubt on the belief in an ultimate free will and primacy of human agency. Neoclassical theory maintains that innovation and price change will always find a way to keep economic growth moving forward. Physicist Geoff West has shown this is not true ("Scale"), that cycles of innovation have their own entropy penalty to them, and these will outrun the ability for more and more rapid innovation to stave off collapse and decay (slides #52 to 58 in this Presentation).

I do have a couple of places where I disagree with Garrett's work; Garrett maintains there is an intimate cause/effect relation between economic inflation and energy return on energy invested. But economic inflation is governed more powerfully by the actions of central banks, who control short term interest rates motivating the large banks to create or reign in money creation via "loans". These decisions can be subject to questionable political decisions or mis-readings of a crisis (e.g. the CoVid recession caused a large over-reaction by the Federal Reserve who then multiplied the money supply; an error we're all paying for today in high inflation). I believe the relation between inflation and EROI is not as close as Garrett maintains. One can imagine that worsening EROI could just be ignored by the money suppliers and so that energy costs would go up, requiring other prices to go down due to unaffordability in a worsening recession. Garrett has speculated in the past (I'm not sure his present thoughts on this) that it is reductions in energy efficiency which cause

recessions. I disagree with this, and in fact it is much easier to see the logic that cause-effect goes the opposite direction, as described above. Recessions can have a variety of causes; disease pandemics, wars, poor policy decisions, bad allocations by markets in search of short term profits... but the resulting recession will affect how money is then allocated, such that improving energy efficiency now takes a back seat to dealing with more pressing needs. However, these points do not affect the validity of the P/W Relation's validity in the data, which was the subject of Hanley's critique. These points only relate to how to interpret what the P/W Relation's origin is, and its ultimate meaning.

My investigations into the P/W Relation are not published except in talks and Presentations I've given. A more compact and newer Presentation on this is here. It's been clear that conventional economists do not like to think of this relation as being true. For one, it says that to reduce energy consumption requires a long term economic recession or depression; a decision to let go of supporting civilization's growth. This is utterly antagonistic to the goal of economists – to promote and guide how to achieve maximum economic growth. There's clearly no hope of getting work such as I've done published in an economics journal. However, my talks have not had any substantive criticisms or refutations of the evidence and reasonings.

Richard Nolthenius – Apr 7, 2025

---

## Author Comment (AC3)

**Response to Tim Garrett**
**by Brian Hanley – Responses in Liberation Sans**

**Table of Contents**

**Preamble – Response to comments from referees regarding "usual format"**

Because of the comments about not following usual format, I think there needs to be a clarification of how this critique paper came about. Originally, it was submitted as a 'Commentary" type of paper. However I was told by editorial that this was too long. So on February 6$^{th}$, editorial recommended it be resubmitted as a "Research" paper.

*Perhaps the best approach would be to change the title, while still making the abstract clear that you are addressing some particular issues/errors in a particular manuscript. You then have the freedom of a research article to explain what that error is (perhaps how it may have arisen) and it's consequences for the original paper and, crucially, it's relevance for future work in this area.*

I accepted that, suggested that I revise the title to say, "Critique" instead of "Commentary", did some revisions and streamlining, and the result is what readers see. There was considerably more material involved than appears here working through various aspects. I apologize for confusion by readers.

A key point I made in that discussion with editorial was this. (**Emphasis added**.)

*This paper is an area that is difficult to enter into, and there is no standard forum for it. The pieces and the whole, the why's and data, are being worked out. Because of this, it is impossible to get funding for, and yet **it appears central to understanding economic limits, and monetary valuation**. Consequently, I want to engage and help nurture it.*

This critique paper is the first time anyone has made a concerted attempt to step through all of the data, independently find sources of data, and validate what was provided in Lotka's Wheel. As such it is a valuable exercise and should be helpful to others. This is a basic step in science – **replication of studies** and comparing results. That is why I named these datasets replicates. The goal of these replicate datasets was to perform sanity checks. I did not expect them to be precisely the same.

Yes, there are issues found. Finding issues, if they exist, is the whole point of a replicated study. If everything checked out, this would be worthy of a commentary note saying so. However, the equations of GGK22 were not actually used as stated, and *w* is a serious error. The *W* dataset provided is based on an error of the variety I would call going to fast. I do not think this is fatal to the concept.

**General remarks**

The critique includes … it is left unstated in the figures what units are used for quantities, and when units in the analyses are presented, they are used inconsistently for the same quantities. There are frequent uses of very large numbers of significant digits for imprecise quantities, up to nine for an exponential growth rate in one case.

**General remarks appear to primarily relate to spreadsheet**

Although not specified, it appears that these remarks refer to the shared spreadsheet's graphs, not the critique paper, as the overly significant digits are only present there, and units are specified on scales in the critique-paper.

**I use Linux  Libreoffice Calc 24.8.1.2 which can have Excel artifacts**

I use Linux almost exclusively, and currently Libreoffice Calc 24.8.1.2. When spreadsheets are accessed using Windows, some figures can have confusing artifacts. *I will make note of this in revision, to warn Microsoft Excel users.*

**Units available**

Units in the spreadsheet are available in the legends, and on the scales, and since this spreadsheet does not refer to any others, if there is a question, the data column can be easily identified. True, the spreadsheet X axes do not say "Year", but anyone familiar with the subject matter should know that. If it is considered important I will add year to the X axes.

**Curve fit defaults are used in spreadsheets and do not matter; these are cosmetic**

Yes, I leave curve fits at the defaults in spreadsheets. I do the same when using mathematical tools like Maple, Matlab, etcetera as well – until reporting them. This is good practice to avoid introducing errors from repeated truncation of precision prior to final reporting, if an equation is ported. I do not see any such "*large numbers of significant digits*" reported in the critique paper with excessive precision.

In any case, if I were to truncate the reporting of the precision in the spreadsheet, this would not affect what the internals of those curve-fitting algorithms do. Internally, the same level of precision would be kept. So at worst, this is a cosmetic matter.

**w* is a constant**

Comment (**emphasis Hanley)**:
A primary conclusion of GGK22 as stated in the abstract is that "In each of the 50 years following 1970 for which reliable data are available, 1 exajoule of world energy was required to sustain each $5.50\pm0.21$ trillion year 2019 US dollars of a global wealth quantity defined as the cumulative inflation-adjusted economic production summed over all history." That is the ratio W for cumulative inflation-adjusted production Y (or $W(t) = 0\ Y(t)\ dt' = \sum_i Y_i$), to the rate of energy consumption R E, is effectively observed to be a constant for a 50-year period since 1970, that is $w = W/E = 0\ t\ Y(t)\ dt' = $ constant. As is argued in the GGK22 article, this constancy, **were it to hold**, it would **imply that inertia, i.e. deep**

**history, plays a guiding role in the evolution of future consumptive needs by civilization**, one far higher than has generally presumed.

The result that w = W /E is a constant was originally anticipated based on thermodynamic reasoning as first described in Garrett (2011) and it was also elaborated upon in GGK22. As with any hypothesis, it's validity rests on a statistical evaluation, as well as the accuracy of the statistics used. In GGK22, a half century period between 1970 and 2019 was used to evaluate w. That the statistic holds for any period before or after **can really only be speculation**, but for that period covering two-thirds of the total historical growth in civilization energy demands w was observed to be nearly constant. **It was speculated that this constancy is a fundamental feature of the global economy**: energy consumption sustains flows along the civilization networks that have previously been built through prior economic production, allowing for fraying of those networks by way of the downward correction from the nominal to the real GDP, as is commonly ascribed to economic inflation.

However, as a time integral of inflation adjusted economic production, the accuracy of the calculation of W – and hence $w = W /E$ – relies on a reconstruction of historical economic data. Any reexamination of the statistics and the methods is important because any reconstruction necessarily requires some subjective choices. It is also challenging. An independent reexamination of the methods used in GGK22 could shed light on better approaches. B. Hanley **provides revised reanalysis datasets from which he makes arguments that $w$ is not in fact a constant, in any time period**.

**...arguments that w is not in fact a constant: Not exactly. I think w is an error**

What I argue is that w is an error, regardless of what statistics are done on some subset of *W*.
Yes, I understand why this is done in Lotka's Wheel. See section 5.1, line 275. I do not think this version of *w* is required to work with the concept.

**Problem with ŵ = Y / (dE/dt)  and Y =  ŵ · (dE/dt) . (dE/dt)=0 causes divide by zero error**

What these two equations say is that when *dE/dt* goes to zero:
*ŵ = Y / (dE/dt)*  is a meaningless expression. *ŵ*  = divide by zero error.
*Y =  ŵ · (dE/dt)* is also a meaningless expression. Divide by zero times zero.

*Though equation 4 (ŵ = Y / (dE/dt) ) generates a divide by zero error, this is used to say real production goes to zero, when divide by zero should mean real Y goes to infinity. This should be seen as indicating an error in previous math.*

*No amount of statistical analysis of the 1971-2019 dataset can justify this use of a constant. It is just an error, and it generates nonsense.*

**Energy**

To address the Hanley critique, I compare the reanalysis dataset in GGK22 to that proposed as alternative of Hanley, **what he terms a replicate dataset**. Statistics for world primary energy consumption since 1970 are stated clearly enough by both sources. Specifically as stated in the Methods of GGK22 Yearly statistics for world primary energy $E_i$ are available for both consumption

and production from the Energy Information Administration (EIA) of the US Department of Energy (DOE) for the period 1980 through 2018 and for consumption from British Petroleum (BP) for the years 1965 through 2019 (DOE, 2020; BP, 2020). A yearly composite of $E_i$ in units of EJ $yr^{-1}$ for the years 1970 to 2019 is created from the average of the three datasets while using single sources when only one is available. The difference between the values in the BP and EIA datasets is significant at 8.5 ± 1.5 %, but it is steady and small relative to the 180 % increase in energy consumption over the 50-year time period considered here.

In the Hanley critique, no argument is presented why the energy data used in GGK22 is problematic. The data set Hanley used was obtained from Our World in Data (Ritchie and Rosado, 2020) which refers to the Statistical Review of World Energy. No averaging is made of consumption and production as in GGK22 but this is a small consideration. Based on that dataset, consumption in 2019 was 588 ExaJoules compared to 595 ExaJoules in the GGK22 dataset. Because this difference is small, it is presumed that the two analyses are in sufficient alignment even if the GGK22 datasets is more comprehensive.

However, Hanley shows a reconstruction of energy demands in 1 CE that is three orders of magnitude smaller than that which may be inferred from GGK22 assuming w is a constant. **Where such large discrepancies exist it should be possible to perform a simple test to determine if one result or the other is unreasonable.** The Hanley critique presents the global consumption rate of energy in 1 CE as 0.0623 ExaJoules, which split among the estimated 225 million people in the world at that time would correspond to a civilization consumption rate averaging to about  9 Watts per person.

Human metabolic needs are approximately 100 Watts. To feed those needs using modern agriculture, the efficiency is about 10% implying order 1000 Watts of consumption.

Presumably ancient agriculture was less efficient, and then of course not all primary energy at that time or any time went into feeding people in a subsistence lifestyle. Militaries, buildings, roads, and sailing networks were also needed. Whatever the source of the calculation, the 9 Watts per person presented by Hanley as part of a superior dataset reconstruction is simply far to small to be plausible, to enable any form of civilization sustenance, and by two to three orders of magnitude. The problems are more of a basic scientific nature when it comes to calculation of cumulative inflation-adjusted world economic production summed over all history W.

**reconstruction ... 1 CE that is three orders of magnitude smaller**

The simple test that makes sense to me is to use per-capita energy in 1800 as the ceiling. This is discussed below.

**Human labor is left out, hunter-gatherers, and slavery was global**

I discuss how human labor is left out of energy consumption (lines 164-165m and lines 351-360). This should be a significant invisible component, that I think factors strongly in the long decline of *Y/E.*

Hunter-gatherers were a large fraction of global population in 1 CE, but what exactly this proportion was is not clear. Recent work has shown the estimation of the roughly 300 societies to be more complex than previously believed[1].

Slavery in Roman times is estimated at 10-20% overall, with 30% of the population enslaved in some regions. For comparison, in the Carolina colony of North America, 60% of the population was slaves. In colonial Boston, 10-12% of the population were slaves.

**Draft animals**

A ratio of one horse equivalent per 10 people may be high for the year 1 CE, particularly globally. North America had none. South America had limited use of lamas and alpacas. In the US, sources conflict, but in 1900, which was quite prosperous, the ratio was about 3 people per horse. In 1720, the ratio may have been much higher, possibly as high as 1 to 1. This declined as land became more expensive and owning horses required buying feed.

**Solar energy is fundamental to agriculture & human intensive can have high yield**

The energy of agriculture comes from the sun, which supplies ~1 KWh $m^2$-day. Plants convert 3%-6% of sunlight energy, which is ~11 KWh per $m^2$-year, or 109 MWh per hectare-year. A big adjustment though is that human intensive agriculture can be more productive per hectare than mechanized methods by a factor of 3 to 20 times. (Multi-cropping can be a factor.) I doubt ancient farmers produced 20 times current mechanized crop yields, but 2-5 times was probably within reach.

**Curation of "wild" lands by human populations**

Europeans, like native American tribes, used to curate their "wild" lands to maximize game and trees that were useful to them. I recently got an education in Northern California tribal methods when spending time on a small ranch that had some "untouched" land. That land had been  cared for with fire to keep grazing available for deer, and brush that favored turkeys and rabbits. Oak trees with preferred high acorn yields had been planted and fertilized with fire baked oyster and mussel shells from the bay region, likely through trade. Pinon pines had been planted, etcetera. Thse kinds of techniques are high yield for the labor expended, and work well as long as the population density isn't too high.

Note that the concept of "interest" comes from the "interest" that the landowner got from animals born on their land.

**Sailing ships**

Yes, it is true that sailing networks had some impact. I don't have a value for energy of sail in the global economy. I didn't find someone that had done that.
* * *
1     Zhu D, Galbraith ED, Reyes-García V, Ciais P. Global hunter-gatherer population densities constrained by influence of seasonality on diet composition. Nat Ecol Evol. 2021 Nov;5(11):1536-1545. doi: 10.1038/s41559-021-01548-3. Epub 2021 Sep 9. PMID: 34504317; PMCID: PMC7611941.

**Concentration on 1 CE low value ignores the high confidence E value of 1800 CE.**

In this sequence from A to C, it should be obvious by inspection that the significant difference is not in the *Y* datasets, regardless of any possible issues. It is the difference in E datasets that generates the huge *Y/E* discrepancy, and by extension the *W/E* discrepancy. Note that the 1800 to present *E* data in figure 1-B are high confidence. It is this E difference that drives the W/E

[Figure]

**Figure 1-B energy datasets – revised based on alternative assumptions**

[Figure]

In this 1-B figure, multiple alternative exponential interpolations are shown. Red upper curve meets the Lotka's Wheel 1 CE value. Green shows interpolation to 100X the *E_Rep* value, and blue shows an interpolation to 10X the *E_Rep* value. In black is shown the interpolation of a steady level of 1800 CE energy per capita.

The Lotka's Wheel *E* dotted curve 1 CE value of 46.172 EJ has to drop by a factor of 2.7 times to get down to the high confidence *E_Rep* 1800 CE value of 20.35. This does not appear reasonable.

Let's try using per-capita energy of 1800 as a guide, and then simply hold per-capita E steady. This should be an overestimate, that can be considered an upper limit.

1800 CE:     20.35 EJ / 953.56 million Pop = 0.021341939 EJ/mPop

  1 CE:      225.82 mPop · 0.021341939 EJ/mPop = 4.82 EJ

Interpolation to 4.8 EJ is shown in yellow, just below the 100X interpolation.

If I accept this overestimate limit of 4.8 EJ in 1 CE, that does not make the problems with *Y/E* and *W/E* go away, it makes them smaller, but not in a way to change my critique significantly.

**Reasonable 1 CE value range: Low of 1.7 *EJ, high of 3.75 EJ**

I can accept raising global energy consumption, but not by a factor of around 750X. Perhaps split the difference between the 4.8 EJ that is obtained by using the 1800 CE energy per capita, and my 1 draft animal per 10 people of ancient Rome, then make the range +/- half the midpoint to ceiling or floor. This would yield, roughly 2.7 EJ globally as the midpoint, with a high of 3.75 EJ and a low of 1.7 EJ.

**Market Exchange Rate (MER) versus Purchasing Power Parity (PPP)**

The approach outlined in the Methods section of GGK22 is that Economic production is tallied and averaged using World Bank (WB) and United Nations (UN) statistics for the years 1970 to 2019 (The World Bank, 2019; United Nations, 2010) and expressed here in units of trillions of market exchange rate, inflation-adjusted "real"-year 2019 dollars. So there are two key points here, which are that the production statistics must be adjusted for inflation and expressed in Market Exchange Rate (MER) values.

The reason to do so was described in greater detail in a prior article referenced in the methods by Garrett et al. (2020) and in the paper the critique references by Garrett (2011). In the Hanley et al. critique, no criticism is made of statistics used or the choice to focus on real, MER dollars so presumably that is not an aspect in contention.

However, Hanley mixes and matches datasets to reconstruct a long time series for economic production that is inconsistent in its choice of units for each year, and so cannot be used for the purpose of addition to calculate W. The economic production statistics they use since 1990 appear to have as their original source the Penn World Tables, although details are not given. I**f this is the case, the both key points of MER, inflation-adjusted dollars are satisfied**.

For data between 1960 and 1990, the Hanley critique states use of data from the Federal Reserve (NYGDPMKTPCDWLD). These are expressed in MER dollars but they do not account for inflation to convert from current-year nominal to fixed-year real dollars. This is explicitly stated in the Federal Reserve dataset but not acknowledged by Hanley. The description in Hanley is difficult to follow but it appears a scaling multiplier is applied to the Penn World Table datasets to match the two economic production datasets post-1990, and that the Federal Reserve data set is used as the basis for a longer time series. Not surprisingly, by not adhering to the requirement of using real 2MER dollars for all years the result is an average 44% difference between the GGK22 and Hanley datasets. The difference is washed away by Hanley with "The discrepancy is due to a change in how OWID GWP and its data sources are calculated versus other datasets, and the difference is consistent enough."

This [Hanley "*The discrepancy is due to a change in how OWID GWP and its data sources are calculated versus other datasets, and the difference is consistent enough.*"] is misleading. For the purpose of calculating W, and error will be cumulative, in this case introduced by not considering the growing devaluation of purchasing power through inflation. Most importantly, for the purpose of calculating W , it makes no sense to sum dollars from one year to the next where they are expressed in the different units of current dollars from different years. The error is like adding 1 pint to 1 quart and saying there are two quarts.

**Take Y_Rep off the table and the replication of results problem persists**

The simple step of taking the Hanley critique *Y_Rep* data off the table and replacing it with *Y_LW* (the Lotka's Wheel dataset) and dividing by the *E_Rep* data, shows that the concerns voiced do not matter for the point being made – that there is a major discrepancy that appears on its face to falsify the *W* dataset provided by GGK22, and falsify the belief that the speculation that the 50 year modern period can simply be extended back into the past is true. There is an effect on the present from the past, but it is more complex.

That said, it is my belief that this thermodynamic approach is so compelling that what is needed is to try to figure out why, rather than throwing out Garrett's fundamental hypothesis.

**Figure 1-C with addition of Y_LW/E_Rep**

[Figure]

**Points addressing the MER vs PPP concerns.**

To perform a proper replication study, I had to find what I considered the most correct dataset independently. This was the Our World in Data (OWID) group's data, which is based on Maddison project and World Bank. This represents the career work of multiple teams that deal with the issues raised by Garrett. My confidence in OWID work is very high.

**Inflation is accounted for in OWID dataset**

Per the OWID documentation, "*This data is adjusted for inflation and for differences in living costs between countries.*"[2]

**Data between 1960 and 1990, using NYGDPMKTPCDWLD is accounted properly**

The purpose of using this NYGDPMKTPCDWLD dataset was to be able to do a better job of interpolating the 10 year intervals than a linear or exponential interpolation.

The dataset provided by OWID has three 10 year intervals from 1960 to 1990. The NYGDPMKTPCDWLD was scaled to fit at each end of the three intervals. This automatically corrected for inflation, because each end was corrected for inflation by definition. Using these scalars, the NYGDPMKTPCDWLD data were used to populate the interval spans.

I needed to do this because without it I would lose significant detail between 1970 and 1990. That would result in a gross mismatch within those intervals when comparing with Garrett's datasets. That this was successful is visible by inspection in figures 1-E and figure 5, from 1970 to 1990. While my *W_Rep* values are higher, they mirror the variation of *W_LW* values.

**PPP vs MER background**

For deep time estimations, one must make use of PPP techniques almost exclusively. Why is this? If one thinks about it, how can an archeologist make use of, say, records from Rome to figure out what people had in terms of money? It is not possible to have a real market exchange rate into currencies that did not exist. What can be done is to find out, for instance, how many denari were paid for some basket of products necessary for living, and luxury products as well. This falls under the umbrella of how PPP is done.

**44% difference between the GGK22 and Hanley datasets**

I assume that the 44% number comes from the mean of the dataset ratios I show in column C of the spreadsheet.

If I use area under the curves, GGK22 $\Sigma Y$ = \$3,380 trillion and Hanley $\Sigma Y$ = \$4,419. The ratio of Hanley/GGK22 = 1.31, or 31% larger. It correlates well enough to the mean of the ratios. This is still close enough, as we saw above in the Figure 1-C curve using GGK22's *Y* dataset.

I do not agree that I intended to "wash away" this difference. I reported the difference, although not in the critique-paper. What I intended to convey is that these datasets are close enough that these quantitative changes will not qualitatively change results in a meaningful way. I show this in the figure 1-C graph above.

Additionally, I already showed that quantitative differences did not make a large qualitative impact on *W* in figure 2 of the critique, reproduced below. Figure 2 also provides a good segue into the next section.

**Equation 5 difficult to understand.**

For the reconstruction of GDP for earlier years, Eq. 5 is difficult to understand and highly contrived given e ln x = x. The reconstruction refers to Our World in Data datasets for GDP statistics pre-1960,
* * *
2    https://ourworldindata.org/grapher/global-gdp-over-the-long-run

which in turn appear to use the same Maddison database used by GGK22. This database appears to have been adjusted for inflation, which is good.

**Overthinking?**

I suspect this puzzlement is due to overthinking it. I simply presented a standard exponential interpolation method. If I had said, "exponential interpolation" this issue might not have arisen. I was just being careful to show exactly how the exponential interpolation was done.

The second equation is just solving the exponential on the left for *a*.

Given that generally speaking, population rose exponentially, and was a primary driver of GDP, I think that an exponential interpolation is required.

**Adjustment from PPP to MER dollars**

However, what is not considered is the second necessary adjustment to convert from purchasing power parity dollars used in the Maddison database to market exchange rate dollars. Specifically, as stated by GGK22: The dataset is adjusted for inflation and to convert from currency expressed in purchasing power parity dollars to market exchange units using as a basis for adjustment the time period between 1970 and 1992 for which concurrent market exchange rate (MER) and purchasing power parity (PPP) statistics are available.

An argument might be made that PPP dollars are preferable to MER dollars. I don't believe this would be appropriate for physical reasons. Whatever the choice, there must be consistency when creating a continuous time series of economic production. The final dataset for Y Hanley uses is a mix of real, inflation-adjusted, MER, and PPP data. Without consistent careful adjustment for inflation and to MER dollars across the full 2000 year time series, the numbers in each year cannot be added to create the central quantity W.

Any summation must use numbers with the same units. Not all dollars are the same.

**PPP to MER is a red herring**

First, I find it hard to believe that it is possible to meaningfully convert PPP to MER in deep time. One might be able to do it to some degree going back 800 years, perhaps. This is as far as Schmelzing got with interest rates[3]. But one has to ask the question, what does MER mean when applied to global numbers?

PPP was created to correct the errors inherent in MER figures. After conversion, without knowing what the MER figures were, how does one reconstruct MER? To reconstruct MER figures from PPP, one would need to have the original MER figures *for each country in that time,* or at least the exchange rates by year. MER to PPP conversions, like exchange rates, are not static. They change by the year. (Reported usually by the month.) It just doesn't make sense to do this, or to think that figures in one century could be applied to figures in another century. I think this is a mistake.
* * *
3 https://www.bankofengland.co.uk/working-paper/2020/eight-centuries-of-global-real-interest-rates-r-g-and-the-suprasecular-decline-1311-2018

In addition to this problem, when PPP approaches are used to estimate GDP worldwide in the first place, how can you differentiate between nations? What would such a conversion even mean?

Since PPP normalizes to the physical real world economy, I think that PPP is the more accurate. I did not criticize the use of MER in modern times, as my primary goal was to perform a sanity check, and globally, MER or PPP are close enough.

**Background on CPI**

Fundamental to the concept of inflation is that to calculate it requires a consumer price index of some kind. This index is a basket of commodities recognized as fundamental to individuals living in a civilization. However, when one commodity inflates too much, the economy substitutes another that is cheaper. Because of this, the concept of substitution is key. What this also means is that inflation has built-in limits.

The US CPI has gone through many changes over the past century[4].

Inflation can (rarely) be very high[5]. From my experience in nations that experienced hyper-inflation, in each of those nations, people turned to other nation's currencies at least to some degree. The US dollar is a common alternative currency, but the Euro, Swiss Franc, and British pound were also present. Essentially, demand evolves a "basket of currencies" response.

Note that in one of the examples, post-WW2 Hungary, the inflation was deliberately engineered to make paying off existing debt cheaper, and encourage entrepreneurship. This strategy was successful, and the hyper-inflation ended. (As is usually the case, this capsule discussion elides significant complexity and context.) I'll stop here going down that rabbit-hole.

**1 CE value of *W**

As a note, the value of W obtained by GGK22 for year 1CE was not a summation of a much longer dataset as stated in Section 1.5.2 of the critique. The method was stated in the Methods section of GGK22 to be something quite different, an inference from the GDP at that time assuming that W and population were growing equally fast. Note too the discussion of uncertainties that might accompany this assumption. The value for cumulative production in 1 CE. W(1) is obtained by assuming that W was growing as fast as population at that time at rate and that Y (1) = R W W (1). Population data from 1 CE and 1 century before and after suggest that global population was 170 million and growing at 0.059 % yr −1 (United States Census Bureau, 2018).

**GGK22 method inferring from GDP that *W* & population grew equally fast is an error**

If $Y_i$ = 0 this means that humans ceased to exist. If humans do not exist all of this is moot.  So, $Y_i$ cannot be zero nor less than zero. For any time in which humans exist, there will be a $Y_i$.
* * *
4   https://www.bls.gov/cpi/additional-resources/historical-changes.htm#
5   https://www.investopedia.com/articles/personal-finance/122915/worst-hyperinflations-history.asp

*W* will always increase. Equation 1 requires that for any non-zero *Y*, *W* must increase each year, because *Y* less than or equal to zero is ruled out.

Because *W* is the summation of all previous $Y_i$, **W will always be larger and faster growing than Y.** *Y* only grows by $Y_i - Y_{i-1}$ while *W* grows by $Y_i$. Proportionally, *W* growth slows down as *i* rises, but the amount by which *W* grows is always greater than the amount by which *Y* grows.

*Y* can decrease YoY continuously, asymptotic to zero, and *W* will continue to increase continuously during this time.

Because the energy humans themselves contribute to the economy is not accounted for in energy accounting today, this is also left out of energy accounting in the past. This should create anomalies in the distant past of high GDP to energy consumption (high *Y/E*).

YoY, *W/E* can decrease, stay the same, or increase.

For *W/E* to decrease YoY, then $E_i/E_{i-1}$ must be a ratio greater than the ratio of $Y_i/W_{i-1}$; for *W/E* to stay the same, then $E_i/E_{i-1}$ must be equal to the ratio of $Y_i/W_{i-1}$; for *W/E* to increase, $E_i/E_{i-1}$ must be less than the ratio of $Y_i/W_{i-1}$.

Therefore, all cases except for *W/E* increasing, must have $E_i/E_{i-1}$ greater than 1.

If *W/E* decreases YoY, this means that YoY *Y/E* decreased.

If *W/E* stays the same YoY, then $E_i/E_{i-1}$ must be greater than 1, and YoY *Y/E* increased.

For $E_i/E_{i-1}$ less than 1, YoY, *W/E* must be increasing and *Y/E* must be increasing by a higher YoY ratio.

**Figure 2 discussion. We can agree GGK22 values of W are not from eq. 1.**

[Figure]

*W curve. A. Log scale. B. Linear scale. The upper black dotted curve is the Lotka ' s W heel supplement W dataset. The lower gray dotted curve is $W_{LW}$ for 1-2019 CE created per equation 1. Solid curve is W Rep for 1-2019 CE, also created per equation 1. What is visible here is that the algorithm of equation 1 when applied to generate W from $Y_{LW}$ and the $Y_{Rep}$, data produces the expected curves that begins with the year 1 CE value. The provided Lotka's Wheel supplement column labeled as W does not.*

We agree that the *W* curve supplement dataset provided by GGK22 was not produced using equation 1 ($W(t) = \Sigma\ Y_j$ ) that was provided.

I think it would be nonsense to say that deeper history does not exist. Thus, in Figure 2 (reproduced just above)the first roughly 500 years is a throw-away. However, as I discuss in lines 51-55, there is not a good formal basis for defining that deep time economic history.

Yes, the method stated in GGK22 says how this value was obtained. I am aware of this. I should have said that the only way to reconcile the *W* curve data with equation 1 is to assume deeper history.

**Claims of misrepresentation of *W* are false.**

While there are inevitable uncertainties in the reconstruction of W as with any other, the yearly values of W since 1970 that are emphasized here cover two-thirds of total growth, so the calculations are more strongly weighted by recent data that are presumably most accurate. Thus, calculation of W, most particularly the conclusion that w is nearly a constant, can be shown to be relatively insensitive to uncertainty in the older statistics (Garrett et al., 2020).
Even the datasets provided in the supplement of GGK22 are misrepresented, although it is unclear how or why. Figure 1E pretends to reproduce the GGK22 data. However, it shows a distinct positive trend in the time series between 1970 and 2019 for w = W /E. This is false. Figure 2c for w in GGK22 shows a different time series with no clear trend. The value presented by Hanley for w in 1970 is about 5 (although no units are given). In the GGK22 dataset provided in the supplement it is 6.1 inflation-adjusted year-2019 MER USD per ExaJoule per year. The difference may seem small but it is significant where constancy is claimed.

**Figure 1-E with GGK22 W dataset displayed – Note fig 5 annotates 1-E**

[Figure]

Here I added the gray dashed curves from the *W* dataset provided in the GGK22 spreadsheet. That data does not quite have a slope of zero, as shown.

Simply put, I cannot explain the *W* data from GGK22. It conflicts with my replication effort. The $W_{LW}$ curve uses equation 1, which was provided as the definition of *W*, and I summed the GGK22 supplied *Y* data.

The data is what it is. I only report it.

**Using GGK22 provided equation 1 to create the *W* dataset is not "pretending".**

Either equation 1 provided in GGK22 is the definition of *W* or it is not. Providing a definition and then ignoring it in favor of a dataset produced in some other way renders any discussion meaningless. All that I have done is to use the definitions provided and implement them. I don't think that can be classified as pretending.

Figure 2 is the only place that I showed the GGK22 *W* dataset – a heavy black dotted curve. Otherwise, I ignored it, because it obviously does not follow the rules defined in GGK22 for producing it.

However, I can add the GGK22 W data to figures 1-E and 5 to highlight the discrepancy if desired.

**Section 2.2 math – the rate growth equation (interest calculation).**

The mathematics in section 2.2 makes no sense. How is economic production Y equal to a growth rate g? How can a dimensioned rate r be added to a dimensionless number unity? It also appears to refer to calculus but not in any way that is familiar or that leads to results that are comprehensible.

**The simplest equation for interest rate growth.**

First, let me point out that this growth equation is an empirically founded confirmation of Garrett's hypothesis that the future is dependent on what comes before it. This equation encapsulates this fundamental empirical fact of capitalism, that the creation of more value depends on the capital one has to do it with. Financiers represent their holdings of capital as units of currency, just as GDP does. For these reasons, this is the most correct equation type to use for behavior of $Y$.

Second, I agree that the GDP value of $Y$ is not directly equal to the growth rate curve itself, unless $Y_0 = 1$. I apologize for misspeaking there. I should have written $Y(t) = Y_0 \cdot g^t$.

The expressed growth rate equation was intended to boil the equation down to its essence, to make the point that an equation of this type needs to be used, and not the average value $w$.

The units are whatever $Y$ is, which for GDP is measured in units of currency.

**Figure: growth equation graph $Y(t) = Y_0 \cdot g^t$**

[Figure]

In this figure, the equation for $Y(t) = Y_0 \cdot g^t$, (where $Y_0 = 1$) is shown for various values of $g$. In the legend, just below each value of $g$ is shown the $1 + r$ components, where $r$ is the growth rate. The value of 1 is a constant that essentially stands for "what came before". This thermodynamic concept of Garrett's is front and center in the concept of interest and yield for investments.

Hopefully this clarifies how this equation works and why it is chosen. To restate, it supports and fits into the thermodynamic concept of growth.

In this figure, the flatline heavy black curve for $Y = Y_0 \cdot 1.0^t$ is the equation for a "steady state" economy where GDP does not change from one year to the next. This is one of a family of equations that come from the $Y = Y_0 \cdot g^t$ equation, because $r$ can conceivably vary in our use, although I do not show this occurring in the above figure.

**For calculus use, the growth equation $Y = Y_0 \cdot g^t$ is needed as the equation for $Y$**

In equation 3 of Lotka's Wheel, which is presented in my section 1.3, Garrett presents an equation for $Y$. My contention is that the reduction of $Y$ to the constant **w** multiplied by $dE/dt$ is an error. Because of this error, a nonsensical result was obtained: that GDP could be any number of currency units and _be reduced to zero by inflation_.

It does not matter if *W* appears linear or not. It does not matter what kind of statistics are done to derive values for *w*. What I am saying is that the correct equation for *Y* must still be used. In Garrett's *dE/dt* = 0, presumed steady-state scenario, it is this equation for *Y* that gives us the *dY/dt* value of zero that one would expect, and everything behaves sensibly.

Using the correct equation, with *g* =1.0, then $Y(t) = Y_0 \cdot 1.0^t$ has a derivative, $Y_0 \cdot 1.0^t \cdot ln(1.0)$, that is zero. And, *dE/dt* does not directly enter into it.

One might expect (as Garrett does in Lotka's Wheel) that both *dE/dt* and *dY/dt* would be zero at the same time, except that the data suggests that when *dE/dt = 0* then *dY/dt* should be low, but still above zero, at least for a while.

**Concluding remarks**

...As a suggested path forward, I suspect that a reconstruction of the timeline of civilization energy consumption in the distant past might be made using a combination of ice-core carbon dioxide concentrations, a carbon cycle model, and an assumption of a mean fuel makeup that does not minimize the widespread reliance on wind, water, and sunlight. People built where they could sail, navigate rivers, and grow crops because they offered forms of primary energy that could be efficiently accessed.
Tim Garrett

I think the suggestions for a path forward that Garrett gives in his concluding remarks are good ones. I also think this should be done in concert with archeologists and others that have wrestled with these issues before.

That said, perhaps we can work out a suggested provisional method for extending into the past prior to 1 CE. Such provisional method(s) would be superseded later, and could be an interesting comparison. If Tim is open to it I could make some suggestions.

---

## Author Comment (AC8)

**Summary and collation of responses to reviewers**

**Brian Hanley**

**04/28/25**

**Table of Contents**

**An unusual situation for a journal in an unusual time**

Ordinarily, journals keep fairly strict boundaries, which makes the forays by Tim Garrett into economics and monetary valuation theory in Earth System Dynamics unusual. And yet, it is justified because atmospheric science has a need to understand how to better model industrial gases $CO_2$ and $N_2O$. Thus we have a novel thermodynamics approach to an economics subject. This approach is valuable.

For editors this presents challenges finding reviewers because these papers are cross-domain. UKRI directs editors to ensure competency of review[1]. viz. *"Improper conduct in peer review of research proposals or results (including manuscripts submitted for publication): this includes ... inadequate disclosure of clearly limited competence"*. In this case, Blair Fix was an important find.

**Table 1: Matrix eval of reviews & Other scientist comments**

| Reviewers | Understanding of Garrett's 2011-2014 thesis (1- 5 scale, 5 is best) | Understanding of Lotka's Wheel critique. (1-5 scale, 5 is best) | Comments useable as specific/applicable to Hanley critique (count) |
|---|---|---|---|
| Garrett | 5 | 4 | 2 |
| Fix* | 3 | 5 | 7 |
| **Other scientists** | | | |
| Jarvis | 3 | NA | 1 |
| Nolthenius | 3 | NA | 1 |

* Previously published in peer reviewed economics journals as first author. I am also first author peer review published in monetary economics, political economics and in climate economics with Garrett – both as a secondary author. I am not an atmospheric scientist, but I have modeled warming using atmospheric science published equations working with Pieter Tans.

**Capsule Summaries**

**Reviewers**

**Timothy Garrett**

I think I can distill the value of Garrett's work by pointing to the hoary Cobb-Douglas equation of economics (which elides energy) and Steve Keen's improvement to this equation, the Energy Based Cobb-Douglas (EBCD) equation. I mention and reproduce a version of EBCD in the introduction. Working with Steve Keen I attempted to test this EBCD equation using real world
* * *
1    UKRI. UKRI Policy and Guidelines on Governance of Good Research Conduct Guidelines. [Formerly Research Councils of the UK (RCUK). Guidelines updated February 2021.] (United Kingdom Research and Innovation, 2013). eprint: https://www.ukri.org/wp-content/uploads/2021/03/UKRI-050321-PolicyGuidelinesGovernanceOfGoodResearchConduct.pdf

data. I was unable to make it work. I cannot say that testing EBCD is impossible, but in the time we had it didn't work. Garrett's work is quite tractable to testing, which makes it falsifiable, and thus, with improvement it may become an important analytic tool.

Since I was the one that had attempted to test EBCD, what was obvious to me just by putting them in juxtaposition, was not to others. I will remedy this in revision.

- Garret did not contest my critiques of inflation going to infinity causing real GDP to go to zero. Nor did anyone else.

- Garrett contested my *E* dataset on the basis that my year 1 energy per capita figure was too low.

  - However, Garrett did not address the insurmountable problem that high confidence *E* data exists back to 1800, and his *E* dataset begins to departs from high confidence *E* records starting in 1969. His year 1 *E* data point is multiples of the high confidence 1800 *E* data for a time with a much lower population. (150-330 million in 1 CE, 1 billion in 1800 CE). His year 1 data point has no significant external support from experts.

  - Discussion and a graph showing estimations of year 1 makes a case for a revised year 1 range. In revision, this will be improved by Ian Morris' energy figures from The Measure of Civilization, per Blair Fix's recommendation.

- Garrett contested my showing *W/E* (*w*) was not constant as misrepresenting his work. He states his opposition as "assuming it is constant...", "can really only be speculation..." and, "It was speculated that..." This does not contest that my data shows that W/E is not constant when using the algorithms GGK22 specified with the GDP data provided by GGK22. Perhaps we can call Garrett's *w* proposition a conjecture. I have no problem changing my title to something like "Testing of Garrett's conjectures" or similar.

  - Only using GGK22's supplement *W* dataset with GGK22's supplement *E* dataset, does one see a constant W/E. I cannot reconcile those by any means, except as Garrett suggests, by assuming it to be true a priori, and conforming to that.

- Although we went around twice about MER vs PPP, whether MER or PPP is correct is moot because I showed that by using GGK22's *pi* adjusted GDP dataset to create a summation, the same problem exists, to wit, that *W/E* is shown to not be constant, and the resulting curve is wildly different.

- Garrett's contested point was that market exchange rate (MER) was correct, and purchasing power parity (PPP) was not. We had two rounds of discussion of this issue. I understand Garrett's point about his methods section creation of the *pi* adjustment to Maddison project's PPP data. Garrett believed he was turning Maddison's PPP data into MER. He did not, because there is no meaning to MER without exchange rate records.

Those cannot exist before the currency was created. The US dollar was created July 6, 1785. So this was an arbitrary adjustment to the Maddison dataset.

- Other miscellaneous points are addressed at some length. One that should be helpful to Garrett is the use of the capital growth equation, which empirically implements dependence on existing capital as the basis for growth. This assumption is fundamental to his overall work.

**Blair Fix**

- Blair Fix understood and agreed with my criticisms. He agreed that MER is not appropriate, as Maddison is PPP data. He asked for improved presentation and made suggestions including color and format.

- Fix expressed skepticism of the concept of global inflation. Archeologists use a PPP type method, and as such may see evidence of price changes in the currency of the time, should records exist. Such records are spotty at best. Fix goes further, pointing out that in the distant past most economic activity was not monetized.

After reading the Ian Morris book Fix recommended, and the Fix paper, I understand the issues better. Estimates of GDP, historical or present day, require assumptions, and often simplifications. Archeologists may use the Geary-Khamis dollar, a theoretical/hypothetical unit equal to 1990 USD purchasing power. Garrett's use of a 1990 USD real dollar baseline suggests he may have been aware of 1990 as a benchmark.

Geary-Khamis dollar use has been related to kilo-calories in the amount of grain purchasable. That model turns the Geary-Khamis dollar into a potential thermodynamic model compatible unit, which should be positive for Garrett's thermodynamic model. This unit is, by definition, a PPP type of estimation, and a way to 'monetize' unmonetized economies.

**Other scientist comments**

**Andrew Jarvis**

Jarvis' comments resulted in significant improvement to understanding of Garrett's thesis and strengthening it.

In response to Jarvis, I spent some time stepping through the math in Garrett's original 2011 paper that introduced his thesis. Doing this uncovered a key issue. The $\lambda$ constant with dimensions of Joules per dollar of GDP, is an aggregate varying with time. It cannot be a constant because this represents all of global industry. I tentatively named it $\Lambda(t)$[J/$]. A version of this issue appears in other elements that are also aggregates that are not constants: $\alpha, \varepsilon$, and $\eta$. To what degree some of these variables can be treated "as if" they were constants for some purposes or periods of time, or else simplified into equations that are straightforward to handle is a question opened by this analysis.

I think this is as far as I could reasonably go toward helping to improve the foundations of Garrett's ambitious and intriguing effort to create new foundations of monetary economics at this time.

**Richard Nolthenius**

Nolthenius appeared to be familiar with Garrett's 2011-2014 work, but did not seem to be aware of notes I posted to Jarvis. He claimed an unpublished confirmation of Garrett's W/E which he termed P/W. I could not critique his claims. His interest is appreciated.

Regarding the specific, numbered comments:

1. Adding the Shadow Economy: Shadow economy is outside of scope. I treat this with a couple of paragraphs.

2. The P/W Constancy over different time scales: The P/W over different time scales discussion is not relevant to my critique of GGK22 for the reasons mentioned above.

3. The PPP vs MER argument: Nolthenius agrees with Garrett on MER vs PPP. So I treated this issue for a third time with more citations, despite it being moot.

4. Inflation measure: This is a note to Garrett. It is irrelevant because the Maddison data is already PPP, and thus by definition inflation corrected.

5. Civilization Obeys Thermodynamic Laws: Other authors with different modeling. I have no argument. It is interesting. This is not applicable here.

6. Thermodynamics and Human Agency: These are comments on neoclassical economics. I generally agree. Not actionable.

The section on criticism of Garrett is not relevant to my critique. I discuss it anyway.

**Garrett Review – Response to Garrett round 1**

**Format of the Garrett section is different due to the number of remarks.**

In this section Hanley's response text appears in Liberation Sans. Virtually all of Garrett's remarks appear here in Liberation Serif, reproduced above the section where they are treated. For extra clarity, there is a vertical bar to the left of quotes from Garrett.

**Garrett general remarks**

The critique includes … it is left unstated in the figures what units are used for quantities, and when units in the analyses are presented, they are used inconsistently for the same quantities. There are frequent uses of very large numbers of significant digits for imprecise quantities, up to nine for an exponential growth rate in one case.

**General remarks appear to primarily relate to spreadsheet**

Although not specified, it appears that these remarks refer to the shared spreadsheet's graphs, not the critique paper, as the overly significant digits are only present there, and units are specified on scales in the critique-paper.

**I use Linux  Libreoffice Calc 24.8.1.2 which can have Excel artifacts**

I use Linux almost exclusively, and currently Libreoffice Calc 24.8.1.2. When spreadsheets are accessed using Windows, some figures can have confusing artifacts. *I will make note of this in revision, to warn Microsoft Excel users.*

**Units available**

Units in the spreadsheet are available in the legends, and on the scales, and since this spreadsheet does not refer to any others, if there is a question, the data column can be easily identified. True, the spreadsheet X axes do not say "Year", but anyone familiar with the subject matter should know that. If it is considered important I will add year to the X axes.

**Curve fit defaults are used in spreadsheets and do not matter; these are cosmetic**

Yes, I leave curve fits at the defaults in spreadsheets. I do the same when using mathematical tools like Maple, Matlab, etcetera as well – until reporting them. This is good practice to avoid introducing errors from repeated truncation of precision prior to final reporting, if an equation is ported. I do not see any such "*large numbers of significant digits*" reported in the critique paper with excessive precision.

In any case, if I were to truncate the reporting of the precision in the spreadsheet, this would not affect what the internals of those curve-fitting algorithms do. Internally, the same level of precision would be kept. So at worst, this is a cosmetic matter.

**$w$ is a constant (Here Garrett speaks of $w$ as a speculation.)**

Comment (**emphasis Hanley)**:

A primary conclusion of GGK22 as stated in the abstract is that "In each of the 50 years following 1970 for which reliable data are available, 1 exajoule of world energy was required to sustain each 5.50±0.21 trillion year 2019 US dollars of a global wealth quantity defined as the cumulative inflation-adjusted economic production summed over all history." That is the ratio W for cumulative inflation-adjusted production Y (or W (t) = 0 Y (t) dt ′ = $\sum$ i $Y_i$ ), to the rate of energy consumption R E, is effectively observed to be a constant for a 50-year period since 1970, that is w = W /E = 0 t Y (t) dt ′ = constant. As is argued in the GGK22 article, this constancy, **were it to hold**, it would **imply that inertia, i.e. deep history, plays a guiding role in the evolution of future consumptive needs by civilization**, one far higher than has generally presumed.

The result that w = W /E is a constant was originally anticipated based on thermodynamic reasoning as first described in Garrett (2011) and it was also elaborated upon in GGK22. As with any hypothesis, it's validity rests on a statistical evaluation, as well as the accuracy of the statistics used. In GGK22, a half century period between 1970 and 2019 was used to evaluate w. That the statistic holds for any period before or after **can really only be speculation**, but for that period covering two-thirds of the total historical growth in civilization energy demands w was observed to be nearly constant. **It was speculated that this constancy is a fundamental feature of the global economy**: energy consumption sustains flows along the civilization networks that have previously been built through prior economic production, allowing for fraying of those networks by way of the downward correction from the nominal to the real GDP, as is commonly ascribed to economic inflation.

However, as a time integral of inflation adjusted economic production, the accuracy of the calculation of W – and hence $w$ = W /E – relies on a reconstruction of historical economic data. Any reexamination of the statistics and the methods is important because any reconstruction necessarily requires some subjective choices. It is also challenging. An independent reexamination of the methods used in GGK22 could shed light on better approaches. B. Hanley **provides revised reanalysis datasets from which he makes arguments that $w$ is not in fact a constant, in any time period**.

**...arguments that $w$ is not in fact a constant: Not exactly. I think $w$ is an error**

What I argue is that $w$ is an error, regardless of what statistics are done on some subset of *W*.

Yes, I understand why this is done in Lotka's Wheel. See section 5.1, line 275. I do not think this version of *w* is required to work with the concept.

***Problem with $\hat{w}$ = Y / (dE/dt) and Y = $\hat{w}$ · (dE/dt) . (dE/dt)=0 causes divide by zero error***

What these two equations say is that when *dE/dt* goes to zero:

$\hat{w}$ = Y / (dE/dt) is a meaningless expression. $\hat{w}$ = divide by zero error.

Y = $\hat{w}$ · (dE/dt) is also a meaningless expression. Divide by zero times zero.

*Though equation 4 (ŵ = Y / (dE/dt) ) generates a divide by zero error, this is used to say real production goes to zero, when divide by zero should mean real Y goes to infinity. This should be seen as indicating an error in previous math.*

*No amount of statistical analysis of the 1971-2019 dataset can justify this use of a constant. It is just an error, and it generates nonsense.*

**Energy**

To address the Hanley critique, I compare the reanalysis dataset in GGK22 to that proposed as alternative of Hanley, **what he terms a replicate dataset**. Statistics for world primary energy consumption since 1970 are stated clearly enough by both sources. Specifically as stated in the Methods of GGK22 Yearly statistics for world primary energy $E_i$ are available for both consumption and production from the Energy Information Administration (EIA) of the US Department of Energy (DOE) for the period 1980 through 2018 and for consumption from British Petroleum (BP) for the years 1965 through 2019 (DOE, 2020; BP, 2020). A yearly composite of $E_i$ in units of EJ yr$^{-1}$ for the years 1970 to 2019 is created from the average of the three datasets while using single sources when only one is available. The difference between the values in the BP and EIA datasets is significant at 8.5 ± 1.5 %, but it is steady and small relative to the 180 % increase in energy consumption over the 50-year time period considered here.

In the Hanley critique, no argument is presented why the energy data used in GGK22 is problematic. The data set Hanley used was obtained from Our World in Data (Ritchie and Rosado, 2020) which refers to the Statistical Review of World Energy. No averaging is made of consumption and production as in GGK22 but this is a small consideration. Based on that dataset, consumption in 2019 was 588 ExaJoules compared to 595 ExaJoules in the GGK22 dataset. Because this difference is small, it is presumed that the two analyses are in sufficient alignment even if the GGK22 datasets is more comprehensive.

However, Hanley shows a reconstruction of energy demands in 1 CE that is three orders of magnitude smaller than that which may be inferred from GGK22 assuming w is a constant. **Where such large discrepancies exist it should be possible to perform a simple test to determine if one result or the other is unreasonable.** The Hanley critique presents the global consumption rate of energy in 1 CE as 0.0623 ExaJoules, which split among the estimated 225 million people in the world at that time would correspond to a civilization consumption rate averaging to about 9 Watts per person.

Human metabolic needs are approximately 100 Watts. To feed those needs using modern agriculture, the efficiency is about 10% implying order 1000 Watts of consumption.

Presumably ancient agriculture was less efficient, and then of course not all primary energy at that time or any time went into feeding people in a subsistence lifestyle. Militaries, buildings, roads, and sailing networks were also needed. Whatever the source of the calculation, the 9 Watts per person presented by Hanley as part of a superior dataset reconstruction is simply far to small to be plausible, to enable any form of civilization sustenance, and by two to three orders of magnitude. The problems are more of a basic scientific nature when it comes to calculation of cumulative inflation-adjusted world economic production summed over all history W.

**reconstruction ... 1 CE that is three orders of magnitude smaller**

The simple test that makes sense to me is to use per-capita energy in 1800 as the ceiling. This is discussed below.

**Human labor is left out, hunter-gatherers, and slavery was global**

I discuss how human labor is left out of energy consumption (lines 164-165m and lines 351-360). This should be a significant invisible component, that I think factors strongly in the long decline of *Y/E.* In modern statistics, human energy is rarely found, although it appears in the Cobb-Douglas function, and Keen's Energy Based Cobb-Douglas revision. Archeologists, however, do account for it back to hunter-gatherers, since Cook's 1971 article[2].

Hunter-gatherers were a large fraction of global population in 1 CE, but what exactly this proportion was is not clear. Recent work has shown the estimation of the roughly 300 societies to be more complex than previously believed[3].

Slavery in Roman times is estimated at 10-20% overall, with 30% of the population enslaved in some regions. For a more modern comparison, in the Carolina colony of North America, 60% of the population was slaves. In colonial Boston, 10-12% of the population were slaves.

**Draft animals**

A ratio of one horse equivalent per 10 people may be high for the year 1 CE, particularly globally. North America had no draft animals in use prior to introduction of horses except for Eskimo sled dogs. South America had limited use of lamas and alpacas. In the US, sources conflict, but in 1900, which was quite prosperous, the ratio was about 3 people per horse. In 1720, the ratio may have been much higher, possibly as high as 1 to 1. This declined as land in the colonies/USA became more expensive and owning horses required buying feed.

**Solar energy is fundamental to agriculture & human intensive can have high yield**

The energy of agriculture comes from the sun, which supplies ~1 KWh m$^2$-day. Plants convert 3%-6% of sunlight energy, which is ~11 KWh per m$^2$-year, or 109 MWh per hectare-year. A big adjustment though is that modern "French intensive" human intensive agriculture can be more productive per hectare than mechanized methods by a factor of 3 to 20 times. (Multi-cropping can be a factor.) I doubt ancient farmers produced 20 times current mechanized crop yields, but 2-5 times was probably within reach.

**Curation of "wild" lands by human populations**

Europeans, like native American tribes, used to curate their "wild" lands to maximize game and trees that were useful to them. I recently got an education in Northern California tribal methods when spending time on a small ranch that had some "untouched" land. That land had been  cared for with fire to keep grazing available for deer, and brush that favored turkeys and rabbits. Oak trees with preferred high acorn yields had been planted and fertilized with fire baked oyster and mussel shells from the bay region, likely through trade. Pinon pines
* * *
2    Earl Cook (1971) The Flow of Energy in an Industrial Society," Scientific American, 225:135-44.
3    Zhu D, Galbraith ED, Reyes-García V, Ciais P. Global hunter-gatherer population densities constrained by influence of seasonality on diet composition. Nat Ecol Evol. 2021 Nov;5(11):1536-1545. doi: 10.1038/s41559-021-01548-3. Epub 2021 Sep 9. PMID: 34504317; PMCID: PMC7611941.

had been planted, etcetera. Thse kinds of techniques are high yield for the labor expended, and work well as long as the population density isn't too high.

Note that the concept of "interest" comes from the "interest" that the landowner got from animals born on their land.

**Sailing ships**

Yes, it is true that sailing networks had some impact. I don't have a value for energy of sail in the global economy. I didn't find someone that had done that.

**Concentration on 1 CE low value ignores the high confidence E value of 1800 CE.**

In this sequence from A to C, it should be obvious by inspection that the significant difference is not in the *Y* datasets, regardless of any possible issues. It is the difference in E datasets that generates the huge *Y/E* discrepancy, and by extension the *W/E* discrepancy. Note that the 1800 to present *E* data in figure 1-B are high confidence, shown as heavy gray. It is this E difference that drives the W/E

[Figure]

Figure 1 of Hanley 2025

**Figure 1-B energy datasets – revised based on alternative assumptions**

[Figure]

In this revision 1-B figure, multiple alternative exponential interpolations are shown. Red upper curve from 1800 meets the Lotka's Wheel 1 CE value. Green shows interpolation to 100X the *E_Rep* value, and blue shows an interpolation to 10X the *E_Rep* value. In black, just below green is shown the interpolation of a steady level of 1800 CE energy per capita.

The Lotka's Wheel *E* dotted curve 1 CE value of 46.172 EJ has to drop by a factor of 2.7 times to get down to the high confidence *E_Rep* 1800 CE value of 20.35, which is a higher population. This does not appear reasonable.

Let's try using per-capita energy of 1800 as a guide, and then simply hold per-capita E steady. This should be an overestimate, that can be considered an upper limit.

1800 CE:    20.35 EJ / 953.56 million Pop = 0.021341939 EJ/mPop

  1 CE:    225.82 mPop · 0.021341939 EJ/mPop = 4.82 EJ

Interpolation to 4.8 EJ is shown in blue, just below the 100X interpolation.

If I accept this overestimate limit of 4.8 EJ in 1 CE, that does not make the problems with *Y/E* and *W/E* go away, it makes them smaller, but not in a way to change my critique significantly.

**Reasonable 1 CE value range: Low of 1.7 EJ, high of 3.75 EJ**

I can accept raising global energy consumption, but not by a factor of around 750X. Perhaps split the difference between the 4.8 EJ that is obtained by using the 1800 CE energy per capita, and my 1 draft animal per 10 people of ancient Rome, then make the range +/- half the midpoint to ceiling or floor. This would yield, roughly 2.7 EJ globally as the midpoint, with a high of 3.75 EJ and a low of 1.7 EJ.

**Market Exchange Rate (MER) versus Purchasing Power Parity (PPP)**

The approach outlined in the Methods section of GGK22 is that Economic production is tallied and averaged using World Bank (WB) and United Nations (UN) statistics for the years 1970 to 2019 (The World Bank, 2019; United Nations, 2010) and expressed here in units of trillions of market exchange rate, inflation-adjusted "real"-year 2019 dollars. So there are two key points here, which are that the

production statistics must be adjusted for inflation and expressed in Market Exchange Rate (MER) values.

The reason to do so was described in greater detail in a prior article referenced in the methods by Garrett et al. (2020) and in the paper the critique references by Garrett (2011). In the Hanley et al. critique, no criticism is made of statistics used or the choice to focus on real, MER dollars so presumably that is not an aspect in contention.

However, Hanley mixes and matches datasets to reconstruct a long time series for economic production that is inconsistent in its choice of units for each year, and so cannot be used for the purpose of addition to calculate W. The economic production statistics they use since 1990 appear to have as their original source the Penn World Tables, although details are not given. I**f this is the case, the both key points of MER, inflation-adjusted dollars are satisfied**.

For data between 1960 and 1990, the Hanley critique states use of data from the Federal Reserve (NYGDPMKTPCDWLD). These are expressed in MER dollars but they do not account for inflation to convert from current-year nominal to fixed-year real dollars. This is explicitly stated in the Federal Reserve dataset but not acknowledged by Hanley. The description in Hanley is difficult to follow but it appears a scaling multiplier is applied to the Penn World Table datasets to match the two economic production datasets post-1990, and that the Federal Reserve data set is used as the basis for a longer time series. Not surprisingly, by not adhering to the requirement of using real 2MER dollars for all years the result is an average 44% difference between the GGK22 and Hanley datasets. The difference is washed away by Hanley with "The discrepancy is due to a change in how OWID GWP and its data sources are calculated versus other datasets, and the difference is consistent enough."

This [Hanley "*The discrepancy is due to a change in how OWID GWP and its data sources are calculated versus other datasets, and the difference is consistent enough*."] is misleading. For the purpose of calculating W, and error will be cumulative, in this case introduced by not considering the growing devaluation of purchasing power through inflation. Most importantly, for the purpose of calculating W , it makes no sense to sum dollars from one year to the next where they are expressed in the different units of current dollars from different years. The error is like adding 1 pint to 1 quart and saying there are two quarts.

**Take **Y_Rep** off the table, and drop the first 500 years, and the replication of results problem persists**

The simple step of taking the Hanley critique *Y_Rep* data off the table and replacing it with *Y_LW* (the Lotka's Wheel dataset) and dividing by the *E_Rep* data, shows that the concerns voiced do not matter for the point being made – that there is a major discrepancy that appears on its face to falsify the *W* dataset provided by GGK22, and falsify the belief that the speculation that the 50 year modern period can simply be extended back into the past is true. There is an effect on the present from the past, but it is more complex.

That said, it is my belief that this thermodynamic approach is so compelling that what is needed is to try to figure out why, rather than throwing out Garrett's fundamental approach.

**Figure 1-C with addition of Y_LW/E_Rep**

[Figure]

**Points addressing the MER vs PPP concerns.**

To perform a proper replication study, I had to find what I considered the most correct dataset independently. This was the Our World in Data (OWID) group's data, which is based on Maddison project and World Bank. This represents the career work of multiple teams that deal with the issues raised by Garrett. My confidence in OWID work is very high.

**Inflation is accounted for in OWID dataset**

Per the OWID documentation, "*This data is adjusted for inflation and for differences in living costs between countries.*"[4]

**Data between 1960 and 1990, using NYGDPMKTPCDWLD is accounted properly**

The purpose of using this NYGDPMKTPCDWLD dataset was to be able to do a better job of interpolating the 10 year intervals than a linear or exponential interpolation.

The dataset provided by OWID has three 10 year intervals from 1960 to 1990. The NYGDPMKTPCDWLD was scaled to fit at each end of the three intervals. This automatically corrected for inflation, because each end was corrected for inflation by definition. Using these scalars, the NYGDPMKTPCDWLD data were used to populate the interval spans.

I needed to do this because without it I would lose significant detail between 1970 and 1990. That would result in a gross mismatch within those intervals when comparing with Garrett's datasets. That this was successful is visible by inspection in figures 1-E and figure 5, from 1970 to 1990. While my *W_Rep* values are higher, they mirror the variation of *W_LW* values.

**PPP vs MER background**

For deep time estimations, one must make use of PPP techniques almost exclusively. Why is this? If one thinks about it, how can an archeologist make use of, say, records from
* * *
4   https://ourworldindata.org/grapher/global-gdp-over-the-long-run

Rome to figure out what people had in terms of money? It is not possible to have a real market exchange rate into currencies that did not exist. What can be done is to find out, for instance, how many denari were paid for some basket of products necessary for living, and luxury products as well. This falls under the umbrella of how PPP is done.

**44% difference between the GGK22 and Hanley datasets**

I do not agree that I intended to "wash away" this difference. I reported the difference, although not in the critique-paper. What I intended to convey is that these datasets are close enough that these quantitative changes will not qualitatively change results in a meaningful way. I show this in the figure 1-C graph above.

I assume that the 44% number comes from the mean of the dataset ratios I show in column C of the spreadsheet.

If I use area under the curves, GGK22 $\Sigma Y$ = $3,380 trillion and Hanley $\Sigma Y$ = $4,419. The ratio of Hanley/GGK22 = 1.31, or 31% larger. It correlates well enough to the mean of the ratios. This is still close enough, as we saw above in the Figure 1-C curve using GGK22's $Y$ dataset.

Additionally, I already showed that quantitative differences did not make a large qualitative impact on $W$ in figure 2 of the critique, reproduced below. Figure 2 also provides a good segue into the next section.

**Equation 5 difficult to understand.**

For the reconstruction of GDP for earlier years, Eq. 5 is difficult to understand and highly contrived given e ln x = x. The reconstruction refers to Our World in Data datasets for GDP statistics pre-1960, which in turn appear to use the same Maddison database used by GGK22. This database appears to have been adjusted for inflation, which is good.

**Overthinking?**

I suspect this puzzlement is due to overthinking it. I simply presented a standard exponential interpolation method. If I had said, "exponential interpolation" this issue might not have arisen. I was just being careful to show exactly how the exponential interpolation was done.

The second equation is just solving the exponential on the left for $a$.

Given that generally speaking, population rose exponentially, and was a primary driver of GDP, I think that an exponential interpolation is required.

**Adjustment from PPP to MER dollars**

However, what is not considered is the second necessary adjustment to convert from purchasing power parity dollars used in the Maddison database to market exchange rate dollars. Specifically, as stated by GGK22: The dataset is adjusted for inflation and to convert from currency expressed in purchasing power parity dollars to market exchange units using as a basis for adjustment the time period between 1970 and 1992 for which concurrent market exchange rate (MER) and purchasing power parity (PPP) statistics are available.

An argument might be made that PPP dollars are preferable to MER dollars. I don't believe this would be appropriate for physical reasons. Whatever the choice, there must be consistency when creating a continuous time series of economic production. The final dataset for Y Hanley uses is a mix of real, inflation-adjusted, MER, and PPP data. Without consistent careful adjustment for inflation and to MER dollars across the full 2000 year time series, the numbers in each year cannot be added to create the central quantity W.

Any summation must use numbers with the same units. Not all dollars are the same.

**PPP to MER is a red herring**

First, I do not believe that it is possible to meaningfully convert PPP to MER in deep time. One might be able to do it to some degree going back 800 years, perhaps. This is as far as Schmelzing got with interest rates[5].  But one has to ask the question, what does MER mean when applied to global numbers?

PPP was created to correct the errors inherent in MER figures. After conversion, without knowing what the MER figures were, how does one reconstruct MER? To reconstruct MER figures from PPP, one would need to have the original MER figures *for each country in that time,* or at least the exchange rates by year. MER to PPP conversions, like exchange rates, are not static. They change by the year. (Reported usually by the month.) It just doesn't make sense to do this, or to think that figures in one century could be applied to figures in another century. I think this is a mistake.

In addition to this problem, when PPP approaches are used to estimate GDP worldwide in the first place, how can one differentiate between nations, to convert all to MER? What would such a conversion even mean?

Since PPP normalizes to the physical real world economy, I think that PPP is the more accurate. I did not criticize the use of MER in modern times, as my primary goal was to perform a sanity check, and globally, MER or PPP are close enough.

**Background on CPI**

Fundamental to the concept of inflation is that to calculate it requires a consumer price index of some kind. This index is a basket of commodities recognized as fundamental to individuals living in a civilization. However, when one commodity inflates too much, the economy substitutes another that is cheaper. Because of this, the concept of substitution is key. What this also means is that inflation has built-in limits.
The US CPI has gone through many changes over the past century[6].

Inflation can (rarely) be very high[7]. From my experience in nations that experienced hyper-inflation, in each of those nations, people turned to other nation's currencies at least to some degree. The US dollar is a common alternative currency, but the Euro, Swiss Franc,
* * *
5    https://www.bankofengland.co.uk/working-paper/2020/eight-centuries-of-global-real-interest-rates-r-g-and-the-suprasecular-decline-1311-2018

6    https://www.bls.gov/cpi/additional-resources/historical-changes.htm#

7     https://www.investopedia.com/articles/personal-finance/122915/worst-hyperinflations-history.asp

and British pound were also present. Essentially, demand evolves a "basket of currencies" response.

Note that in one of the examples, post-WW2 Hungary, the inflation was deliberately engineered to make paying off existing debt cheaper, and encourage entrepreneurship. This strategy was successful, and the hyper-inflation ended. (As is usually the case, this capsule discussion elides significant complexity and context.) I'll stop here going down that rabbit-hole.

**1 CE value of *W**

As a note, the value of W obtained by GGK22 for year 1 CE was not a summation of a much longer dataset as stated in Section 1.5.2 of the critique. The method was stated in the Methods section of GGK22 to be something quite different, an inference from the GDP at that time assuming that W and population were growing equally fast. Note too the discussion of uncertainties that might accompany this assumption. The value for cumulative production in 1 CE. W(1) is obtained by assuming that W was growing as fast as population at that time at rate and that Y (1) = R W W (1). Population data from 1 CE and 1 century before and after suggest that global population was 170 million and growing at 0.059 % yr −1 (United States Census Bureau, 2018).

**GGK22 method inferring from GDP that *W* & population grew equally fast is an error**

Here I go through a proof development for a specific point. Overall, I suspect this can be mooted by use of Morris' dataset.

If $Y_i$ = 0 this means that humans ceased to exist. If humans do not exist all of this is moot.  So, $Y_i$ cannot be zero nor less than zero. For any time in which humans exist, there will be a $Y_i$.

*W* will always increase. Equation 1 requires that for any non-zero *Y, W* must increase each year, because *Y* less than or equal to zero is ruled out.

Because *W* is the summation of all previous $Y_i$, **W will always be larger and faster growing than Y**.  Y only grows by $Y_i - Y_{i-1}$ while *W* grows by $Y_i$. Proportionally, *W* growth slows down as *i* rises, but the amount by which *W* grows is always greater than the amount by which *Y* grows.

*Y* can decrease YoY continuously, asymptotic to zero, and *W* will continue to increase continuously during this time.

Because the energy humans themselves contribute to the economy is not accounted for in energy accounting today, whether to leave it out in the ancient past, for our purposes needs discussion. I will have to see if use of Morris' archeologist produced datasets can be reconciled with overlapping time in high confidence datasets.

YoY, *W/E* can decrease, stay the same, or increase.

For *W/E* to decrease YoY, then $E_i/E_{i-1}$ must be a ratio greater than the ratio of $Y_i/W_{i-1}$; for *W/E* to stay the same, then $E_i/E_{i-1}$ must be equal to the ratio of $Y_i/W_{i-1}$; for *W/E* to increase, $E_i/E_{i-1}$ must be less than the ratio of $Y_i/W_{i-1}$.

Therefore, all cases except for *W/E* increasing, must have $E_i/E_{i-1}$ greater than 1.

If *W/E* decreases YoY, this means that YoY *Y/E* decreased.

If *W/E* stays the same YoY, then $E_i/E_{i-1}$ must be greater than 1, and YoY *Y/E* increased.

For $E_i/E_{i-1}$ less than 1, YoY, *W/E* must be increasing and *Y/E* must be increasing by a higher YoY ratio.

**Figure 2 discussion. We can agree GGK22 values of W are not from eq. 1.**

[Figure]

*W curve. A. Log scale. B. Linear scale. The upper black dotted curve is the Lotka's W heel supplement W dataset. The lower gray dotted curve is $W_{LW}$ for 1-2019 CE created per equation 1. Solid curve is W Rep for 1-2019 CE, also created per equation 1. What is visible here is that the algorithm of equation 1 when applied to generate W from $Y_{LW}$ and the $Y_{Rep}$, data produces the expected curves that begins with the year 1 CE value. The provided Lotka's Wheel supplement column labeled as W does not.*

We agree that the *W* curve supplement dataset provided by GGK22 was not produced using equation 1 ($W(t) = \Sigma\ Yj\ $) that was provided.

I think it would be nonsense to say that deeper history does not exist. Thus, in Figure 2 (reproduced just above)the first roughly 500 years is a throw-away. However, as I discuss in lines 51-55, there is not a good formal basis for defining that deep time economic history.

Yes, the method stated in GGK22 says how this value was obtained. I am aware of this. I should have said that the only way to reconcile the *W* curve data with equation 1 is to assume deeper history.

As said at the outset, the Morris' 16000 year datasets pointed out by Fix may moot this issue.

**Claims of misrepresentation of *W* are false.**

While there are inevitable uncertainties in the reconstruction of W as with any other, the yearly values of W since 1970 that are emphasized here cover two-thirds of total growth, so the calculations are more strongly weighted by recent data that are presumably most accurate. Thus, calculation of W, most particularly the conclusion that w is nearly a constant, can be shown to be relatively insensitive to uncertainty in the older statistics (Garrett et al., 2020).

Even the datasets provided in the supplement of GGK22 are misrepresented, although it is unclear how or why. Figure 1E pretends to reproduce the GGK22 data. However, it shows a distinct positive trend in the time series between 1970 and 2019 for w = W /E. This is false. Figure 2c for w in GGK22 shows a different time series with no clear trend. The value presented by Hanley for w in 1970 is about 5 (although no units are given). In the GGK22 dataset provided in the supplement it is 6.1 inflation-adjusted year-2019 MER USD per ExaJoule per year. The difference may seem small but it is significant where constancy is claimed.

**Figure 1-E with GGK22 W dataset displayed – Note fig 5 annotates 1-E**

[Figure]

Here I added the gray dashed curves from the *W* dataset provided in the GGK22 spreadsheet. That data does not quite have a slope of zero, as shown.

Simply put, I cannot explain the *W* data from GGK22. It conflicts with my replication effort. The $W_{LW}$ curve uses equation 1, which was provided as the definition of *W*, and I summed the GGK22 supplied *Y* data.

The data is what it is. I only report it.

**Using GGK22 provided equation 1 to create the *W* dataset is not "pretending".**

Either equation 1 provided in GGK22 is the definition of *W* or it is not. Providing a definition and then ignoring it in favor of a dataset produced in some other way renders any discussion meaningless. All that I have done is to use the definitions provided and implement them. I don't think  that can be classified as pretending.

Figure 2 is the only place that I showed the GGK22 *W* dataset – a heavy black dotted curve. Otherwise, I ignored it, because it obviously does not follow the rules defined in GGK22 for producing it.

However, I can add the GGK22 W data to figures 1-E and 5 to highlight the discrepancy if desired.

**Section 2.2 math – the rate growth equation (interest calculation).**

The mathematics in section 2.2 makes no sense. How is economic production Y equal to a growth rate g? How can a dimensioned rate r be added to a dimensionless number unity? It also appears to refer to calculus but not in any way that is familiar or that leads to results that are comprehensible.

**The simplest equation for interest rate growth.**

I apologize for misspeaking there. I should have written $Y(t) = Y_0 \cdot g^t$.

First, let me point out that this growth equation is an empirically founded **confirmation of Garrett's overall hypothesis** that the future is dependent on what comes before it. This equation encapsulates this fundamental empirical fact of capitalism, that the creation of more value depends on the capital one has to do it with. Financiers represent their holdings of capital as units of currency, just as GDP does. For these reasons, this is the most correct equation type to use for behavior of Y.

Second, I agree that the GDP value of $Y$ is not directly equal to the growth rate curve itself, unless $Y_0 = 1$.

The expressed growth rate equation was intended to boil the equation down to its essence, to make the point that an equation of this type needs to be used, and not the average value $w$.

The units are whatever $Y$ is, which for GDP is measured in units of currency.

**Figure: growth equation graph $Y(t) = Y_0 \cdot g^t$**

[Figure]

In this figure, the equation for $Y(t) = Y_0 \cdot g^t$, (where $Y_0 = 1$) is shown for various values of $g$. In the legend, just below each value of $g$ is shown the $1+ r$ components, where $r$ is the growth rate. The value of 1 is a constant that essentially stands for "what came before".  This thermodynamic concept of Garrett's is front and center in the concept of interest and yield for investments.

Hopefully this clarifies how this equation works and why it is chosen. To restate, it supports and fits into the thermodynamic concept of growth.

In this figure, the flatline heavy black curve for $Y = Y_0 \cdot 1.0^t$ is the equation for a "steady state" economy where GDP does not change from one year to the next. This is one of a family

of equations that come from the $Y = Y_0 \cdot g^t$ equation, because $r$ can conceivably vary in our use, although I do not show this occurring in the above figure.

**For calculus use, the growth equation $Y = Y_0 \cdot g^t$ is needed as the equation for Y**

In equation 3 of Lotka's Wheel, which is presented in my section 1.3, Garrett presents an equation for $Y$. My contention is that the reduction of $Y$ to the constant **w** multiplied by *dE/dt* is an error. Because of this error, a nonsensical result was obtained: that GDP could be any number of currency units and *be reduced to zero by inflation*.

It does not matter if *W* appears linear or not. It does not matter what kind of statistics are done to derive values for *w*. What I am saying is that the correct equation for $Y$ must still be used. In Garrett's *dE/dt* = 0, presumed steady-state scenario, it is this equation for $Y$ that gives us the *dY/dt* value of zero that one would expect, and everything behaves sensibly.

Using the correct equation, with $g$ =1.0, then $Y(t) = Y_0 \cdot 1.0^t$ has a derivative, $Y_0 \cdot 1.0^t \cdot ln(1.0)$, that is zero. And, *dE/dt* does not directly enter into it.

One might expect (as Garrett does in Lotka's Wheel) that both *dE/dt* and *dY/dt* would be zero at the same time, except that the data suggests that when *dE/dt* = 0 then *dY/dt* should be low, but still above zero, at least for a while.

**Concluding remarks**

...As a suggested path forward, I suspect that a reconstruction of the timeline of civilization energy consumption in the distant past might be made using a combination of ice-core carbon dioxide concentrations, a carbon cycle model, and an assumption of a mean fuel makeup that does not minimize the widespread reliance on wind, water, and sunlight. People built where they could sail, navigate rivers, and grow crops because they offered forms of primary energy that could be efficiently accessed.
Tim Garrett

I think the suggestions for a path forward that Garrett gives in his concluding remarks are good ones. I also think this should be done in concert with archeologists and others that have wrestled with these issues before.

That said, perhaps we can work out a suggested provisional method for extending into the past prior to 1 CE. Such provisional method(s) would be superseded later, and could be an interesting comparison. If Tim is open to it I could make some suggestions.

**Garrett Review – Response to Garrett round 2**

**Overall reply – Consensus between us on most matters of Hanley's critique**

I appreciate that Garrett appears accepting of consensus on most matters of the critique. It is my hope it should be clear that I do not consider this critique to mean that the method of applying thermodynamic modeling to economics is unusable. And yet, I get the sense that we are stuck in a bit of a pothole here. The reason that I (Hanley) performed the work of this critique was to better understand Garrett's work, find any issues, and by doing so strengthen the work. I also hope to make it more accessible to others. It would be quite surprising if everything was perfect considering how far afield from economics atmospheric physics is.

One could perhaps make the case that the economics field is even further from atmospheric physics.

**Garrett discussion of misrepresentation and MER vs PPP**

Reviewers Hanley, Fix, and Jarvis each misrepresent the methods employed in Garrett et al. because they did not considered in its entirety the Methods section of the paper in question. Within that relatively brief Methods section was reference to a more detailed description of how cumulative production was calculated contained in Garrett et al. (2020). In any critique, it should be incumbent that it the methods being criticized be accurately represented, as well as the motivation used.

As a more general point, there seems to have been inadequate attention paid in the comments to the issue of how to add dimensioned quantities. For the question of adding rates of energy consumption it is clear enough, and there appears to be consensus. A rate defined in units of energy per time cannot mix e.g. exajoules per year with millions of barrels of oil equivalent per month. For questions of production and cumulative production, less care was taken. The essence of the calculation of the quantity of cumulative production W (units currency) is that it is a time integral of real, (mostly downward) inflation-adjusted Market Exchange Rate (MER) production (units currency per time). The statistics for production are provided not as a continuous function but in discrete units of currency per year. The challenge is that the units of currency in the sources provided are sometimes not inflation-adjusted (i.e. nominal rather than real) and sometimes in purchasing power parity (PPP) units rather than MER units. Further there is the question for a cumulative quantity of how to initialize the integral and when.

What is essential when adding is that the units be consistent. Production in nominal units from one year cannot be added to production in nominal units from another year without first adjusting for inflation. PPP dollars cannot be added to MER dollars. It's not even clear that PPP dollars for one country can be added to those from another country since the people are different and an adjustment was made for their respective standards of living.

Hanley and Fix in particular miss these points, in particular the methods used. Hanley further misses the approach taken in Garrett et al. with respect to the initialization in 1 C.E. The methods section clarifies what was actually done in Garrett et al. (2022). The reference contained therein to Garrett et al. (2020) is repeated here. The datasets used in Garrett et al. (2022) are modestly different, but the essential approach is the same. There is plenty to reasonably raise questions about within the methods that were adopted without misrepresenting it.

Calculation of cumulative production (from Garrett et al., 2020)

Market exchange rate estimates of Yi, inflation-adjusted to "real" constant year 2010 dollars, are available from the World Bank and the United Nations for the years between 1970 and 2017 (The World Bank, 2019; UNs, 2020). Estimates of real GDP adjusted for purchasing power parity (PPP) 1990 USD are available for each year between 1950 and 1992, and in larger intervals extending back to 1 CE (Maddison, 2003). To calculate W these estimates are converted to market exchange rate MER inflation-adjusted 2010 values. For the time period between 1970 and 1992 for which

concurrent MER and PPP statistics are available, the mean inflation-adjusted ratio PPP/MER is 1.205 with no clear trend.

A historical reconstruction of the annual global GDP is obtained by applying κ(t) to the Maddison PPP values between 1 C.E. and 1970 C.E, applying a cubic spline between sparse data points to obtain annual values, and using World Bank statistics for more recent years (The World Bank, 12019). The value of world cumulative production W is then

$$W(t) = W(1) + \sum_{1}^{t} Y(t) \qquad\qquad (1)$$

where W (1) refers to total accumulated world cumulative production to date in 1 C.E. To obtain a value for W (1), it is assumed that W and world population grew equally fast at that time. Available statistics suggest a population in ca. 1 C.E. (United States Census United States Census Bureau, 2021) that was 170 million and growing by 10 million every hundred years, at a rate of $\eta\, pop = 0.059$ % per year. The estimated value for the real MER GDP in 1 C.E. is 0.147 trillion 2010 USD. Assuming that civilization population and wealth grew at the same rate, i.e., $\eta\, pop = \eta\, W$ , then from Eq. 19 it follows that W (1) = 250 trillion 2010 USD.

One criticism might be that MER dollars should be adjusted to PPP dollars (Cullenward et al., 2011) since market exchange rates fail to account for differences in how people in different countries value equivalent baskets of goods. One rebuttal has been that such equivalents do not exist because different cultures value goods differently and that any discrepancies tend to diminish over time with a half life of three to five years due to the pressures of international and domestic trade (Rogoff , 1996). In the case of the work here, there is another counter-argument which is that there is no intent to address short-term inequalities between nations, only the global sum of all of civilization and its evolution over they long-run. Effectively, there is only one "basket of goods", and that is humanity taken as a whole, including all its social and physical networks.
Rates of global primary energy consumption from all sources E are available from the U.S. Department of Energy (DOE) Energy Information Administration (EIA) for the time period 1980 to 2016 and from British Petroleum between 1965 and 2017 (DOE, 2011; Bri, 2018). Rates of global primary energy consumption and production provided by the EIA have a mean ratio of 99.83% so here it is assumed that the two are equivalent.

**Reply to Garrett's MER and W discussion**

Here, Garrett reiterates claims that his work has been misrepresented. He concentrates primarily on the Market Exchange Rate (MER) vs Purchasing Power Parity (PPP) issue, implicitly suggesting that this may refute the critique points relative to replicated GDP and *W* datasets, and is a highly significant error. Garrett uses an energy metaphor, "*A rate defined in units of energy per time cannot mix e.g. exajoules per year with millions of barrels of oil equivalent per month.*" This metaphor is a bit exaggerated, as it would suggest that Hanley had done something as foolish as adding dollars, Yuan, Yen, Florins, Guilders, Denari, etcetera, between time periods without conversions. I did not make any such error(s). I took a dataset created by others who are accepted experts in the field of economics in deep time.

The Maddisson dataset is a major part of their life's work[8]. I used PPP because those experts in that field used PPP, and because I understand that PPP is the correct choice.

Thus, the point is now made twice regarding MER vs PPP. However, this point shows a misunderstanding of concept by Garrett, as is explained in my previous response. MER is meaningless in the not so distant past, to say nothing of millennia. How does one determine a rate of exchange between US dollars and Denari which ceased to be used 1000 years ago? It is done using the methods PPP uses. In any case, this MER matter is moot because I demonstrate the illustrated problem *without* reference to my replicate GWP data using Maddison Project data. The difference is minimal.

Perhaps Garrett missed that section because of the format, in which I place his text at the top of each section in Times New Roman, and my replies in Serif Sans? There is a lot to read, and people are busy. Things get missed.

If not, and consensus is being avoided because of concerns over MER vs PPP, the fact that the MER vs PPP matter is moot should mean we come to consensus that way. We should achieve consensus because the issue presented in my critique remains clearly visible without any reference to the replication dataset that I created. This is seen under the "**Figure 1-C with addition of Y_LW/E_Rep**" heading. This renders the rest of the discussion moot relative the the primary matter displayed.

The MER vs PPP issue is discussed starting on page 7 of Hanley's "**Response to Tim Garrett**" posted above. The section title is "**Market Exchange Rate (MER) versus Purchasing Power Parity (PPP)**". The Table of Contents displays the headings, that in themselves tell the story, reproduced below.
* * *
8   Reading Ian Morris' book, The Measure of Civilization, has improved my understanding of how the Maddisson numbers were estimated, and the painstaking work involved by archeologists and specialist economists. This edificiation supports the use of PPP, and not MER.

Hanley's methods of interpolation are slightly different. I think those interpolation methods are mildly better, albeit with the caveat that any such interpolation has error margins. In any case, those interpolation methods do not have significant bearing on the issues discussed.

**Additional note**

I do understand the method used by Garrett in GGK22. "mean inflation-adjusted ratio PPP/MER is 1.205 with no clear trend." This value, termed *pi* is, indeed, a ratio between market exchange value and purchasing power parity. It just does not have the meaning Garrett thinks it does when projected back. It is meaningless applied to the past before dollars existed, or where there are no meaningful records of exchange rate. I was consistent in use of PPP values.

See discussion in response to Nolthenius for references on MER vs PPP and when each is appropriate.

**Fix Review – Response to Fix**

**The big picture**

Great references. I will beef up the introduction with material on the difficulty of assessing GDP, and how PPP is required for deep time datasets.

I think that the audacious attempt Garrett engaged to try to understand the relationship between energy consumed to operate economies, the production of that economy, and the valuation of money by what it can buy, is unique and valuable.

There are, indeed, major issues to wrestle with that will always be there. What the components of capital ($K$) are, and how to assess capital's monetary and real economy value is at least as difficult as those of GDP.

I agree that whether the concept of global inflation is meaningful is matter to wrestle with. I lean toward believing that inflation probably is globally meaningful because interest and inflation are integral to the operation of financial systems. Each monetary system, taken alone, usually displays inflation as a characteristic, although the CPI on which inflation is based is crucial for understanding this. Financial systems break when deflation rules, because the greater the deflation rate the greater the disincentive to invest or spend. Inflation can be thought of as the near universal implementation of the demurrage currency of that inspired the Worgle experiment in Austria. The difference is that demurrage cost for holding uninvested currency is fixed and predictable, but inflation is variable and uncertain.

**Comments on format**

1. Charts. Yes to everything. I specifically removed color to comply with colorblindness accessibility standards. But I can put it back and continue to comply.

2. Methods. Either an Appendix, or a supplement works for me. OWID has publications I can try to summarize.

3. Paper order. Ok. I will work on that. I ordered it from the abstract, and I can make that more clear.

**Other datasets of interest**

Ian Morris – The Measure of Civilization. Thanks very much for that cite. The book was ordered and read. I definitely will include. Did not know about him.

**Regarding Garrett's critique**

I will beef up my discussion of Market Exchange Rate vs purchasing power parity (PPP). You can justify PPP into the past and deep past based on a CPI basket of commodities for living.  Market Exchange Rate (MER) has no meaning before a currency existed.

*...So it would seem like if Garrett is recommending MER GDP for recent data, that amounts to switching methods mid-way.*

Yes, exactly. It's switching methods for the last 10%, and doing an operation using this *pi* factor that is not valid to do. Even so, this has a minor effect.

**Jarvis comments – Response to Jarvis: Garrett 2011, equations rendered for economist use**

Here I present an examination of Garrett's 2011 paper that is the earliest publication in the set of articles discussing the relationship of energy consumption to the summation of gross world product throughout human history.

> **Garrett 2011**
> Garrett, T.J. Are there basic physical constraints on future anthropogenic emissions of carbon dioxide?. Climatic Change 104, 437–455 (2011). https://doi.org/10.1007/s10584-009-9717-9

**Key conceptual issue: $\lambda$ is an aggregate $\Lambda(t)$[J/\$] $\equiv \sum \lambda_i(t)$[J/\$] & model taken from snowflake/droplet growth**

The thermodynamic model in Garrett 2011 appendix A, is borrowed from growth of a snowflake or droplet. This atmospheric science model is then used for the human child growing up metaphor (Garrett 2011 Pg 440 after eq 3). In both cases, the model works well enough, as both snowflakes and human children grow in more or less equivalent units by a mechanism that can be considered singular. However, an economy is like a global ecology, with many different organisms interacting. Extending to manufacturing and supply chains, etcetera, within the ecology makes for more complexity.

This simple model appears to have led to a conceptual issue, because snowflakes and droplets have one unchanging growth mechanism. This conceptual issue includes the incorrect belief that if $\lambda$ is not a constant then the thermodynamic model is false. (Garrett 2011 section 4, pg 443) Let us not throw the baby out with the bathwater, because of this conceptual issue.

The $\lambda$ constant has dimensions of [J/\$] (Joules per dollar of GDP). So $\lambda$ is how energy is used to convert resources into utility economy products which are each evaluated using currency for GDP. A bit of thought makes clear that this is industrial production, and it must be an aggregation function. This step of conversion of materials into products incorporates a huge population of different industrial processes with widely varying efficiency of conversion. This population of industrial processes change yearly, and even day by day as do the products produced[9]. Entirely new products having value appear, and old products disappear. So $\lambda$ should represent a large aggregation of equation outputs varying in $t$. This aggregate of functions will have new functions introduced and old ones removed. This necessarily means that $\lambda$ cannot be a constant. It also means that any derivatives are massively complex, because they are the summation of an array of derivatives. This could only be tractable by numeric methods. Any such summation is an estimate.

For clarity, let us represent $\lambda$ with a capital $\Lambda$:
$\Lambda(t)$[J/\$] $\equiv \sum \lambda_i(t)$[J/\$] **where $\lambda_i$ is one of the 359 million plus businesses in the world** [10],
**and each $\lambda_i$ is represented by some unique production function**
* * *
9    In an automobile plant I automated, Ford Motor Company estimated over 100 engineering changes to the vehicles per day made it into the assembly line. Model year conversion was not included in this.

10   Dyvik, EH (2024)  Estimated 359 million companies worldwide in 2023, increased from 2020's 328 million companies. Peak of global numbers in the provided time period. https://www.statista.com/study/102571/companies-worldwide/

**Energy & production equations**

These are shown with their dimensions in brackets. J = Joules, $ = US dollars,

$\Delta G \equiv E_G(t)$ **[J]**  the Gibbs free energy yield from consumption of some fuel source.  (Eq. 1)
$a \equiv E_a(t)$ **[J]**  $= \alpha \cdot E_G(t)$  base energy, for instance electrical power, or torque from an ICE engine.  (Eq. 2)
$w$ **[J]**  $\equiv E_x(t)$ **[J]** $= \varepsilon \cdot E_a(t) = \varepsilon \cdot \alpha \cdot E_G(t)$  *the net exergy*
$\alpha \equiv E_a(t)/E_G(t)$  $= \alpha(t)$ where $0 \leq \alpha(t) \leq 1$
$\varepsilon \equiv E_x(t)/E_a(t) = \varepsilon(t)$ where $0 \leq \varepsilon(t) \leq 1$

$P(t) \equiv Y(t) = E_a(t)[J]/\Lambda(t)[J/\$]$

**Constants that really aren't constant**

$\varepsilon$ and $\alpha$ are dimensionless, treated as constants. However, both are aggregates, and what makes up each efficiency factor is different. Still, these values are between zero and one.

**Alpha ($\alpha$) — $\alpha(t)$ The efficiency of engines, and mixtures of them**

$\alpha$ describes the efficiency of engines used to harness energy for human use, which can vary based on maintenance[11], type of generation equipment[12] and fuel source. This efficiency is shown in figures A and B, below. Note that the efficiencies shown are maximums for top of the line, properly tuned engines. In the real world, each engine of the same type is a bit different, equipment ages, and the mix of power sources can vary a great deal during the course of a day, to say nothing of a year or from year to year. Thus, in the real world, $\alpha$ should be a function varying in $t$.

$\alpha \equiv E_a(t)/E_G(t) = \alpha(t)$

Can $\alpha$ be treated as a constant?
I conjecture $\alpha$ may be defensible as a constant over short periods.  See figures A and B, below.
* * *
11  For instance, a gas turbine's efficiency varies by model, how it is maintained, RPM, and true utilization factor of the output.
12   Smil, Vaclav (2018) Energy and Civilization: A History, The MIT Press.
https://mitpress.mit.edu/9780262536165/energy-and-civilization/
https://visualizingenergy.org/maximum-efficiencies-of-engines-and-turbines-1700-2000/

[Figure]

*Figure A: Watts of capacity at maximum efficiencies, by year and type. Smil, V. Figures from Smil's data do not include solar, wind, or nuclear energy. This figure is provided to make a point about aggregate energy.*

[Figure]

*Figure B: Efficiency: Maximum percent of $E_G(t)$. Smil, V. Figures from Smil do not include solar, wind, or nuclear energy. This figure is provided to make a point about aggregate energy.*

**Epsilon ($\varepsilon$) — $\varepsilon(t)$ the exergy used to manufacture products, and variable mixtures of them**

$\varepsilon$ is a different aggregate that to my knowledge is not tracked. $\varepsilon$ gives the fraction of $E_a(t)$ that ends up being used to make products and is not wasted (mostly as heat). This means that $\varepsilon$ should be different for each $\lambda_i$ and over the aggregation of $\Lambda(t)$ probably display a broad range. Because that $\Lambda(t)$ mix varies during the course of any given year and from year to year, I think it is safe to say that $\varepsilon$ is a function in $t$.

$$\varepsilon \equiv E_x(t)/E_a(t) = \varepsilon(t)$$

That said, I think this can be ignored because there is no data to work with. Thus, I would normally set $\varepsilon = 1$, and use $\alpha(t)$.

In the real world, $E_a(t)$ is the majority of what is available as energy data. Electricity generation is easy to obtain. $E_G(t)$ data would come from consumption of fossil fuel primarily, with some contribution from nuclear. $E_a(t)$ easy, and $E_G(t)$ mostly is a theoretical issue not normally used in economics, except that it is desirable to have the highest possible efficiency conversion of $E_G(t)$ to $E_a(t)$.

What the $E_G(t)$ for solar and wind would be is an interesting question. The first thing that comes to mind is the energy required to manufacture, transport, install, and maintain solar and wind over their useful life. However, this amortized cost is not part of the calculation for $E_G(t)$ in any other case.

**Eta ($\eta$) and natural logarithms of energy functions**

Garrett 2011 makes use of log functions and their derivatives, apparently with the assumption that these growth equations are logarithmic. However, I do not think this assumption being always true is warranted, although in economics, endless exponential growth equations are postulated and used. Additionally, $\eta$ is a function that varies in $t$ per the definition given by Garrett.

$$d \ln(a)/dt = d \ln(\Delta G)/dt = \eta \qquad \text{Garrett 2011, end of figure 1 caption.}$$

This means two different energy equations have the same slope always at any time $t$. This could happen if and only if, the equations are the same equation, but modified by adding or subtracting a constant. This would mean $a \equiv F_1(t)$ and $\Delta G \equiv F_2(t)$.

The addition of a constant to $t$ will shift the curve left or right. [ $F_1(t+C) = F_2(t+C)$ ] The addition of a constant to the equation will shift the curve up or down. [ $F_1(t) + C = F_2(t) + C.$ ] In these cases, the first derivative does not change from the constant. In the first, the constant C is added outside of the function, as $t$ can be any value. In the second, the C added to the equation disappears in the derivative.

[Figure]

$$E(t) = \frac{\left(Sx1 \cdot Sk1 \cdot Shak500 \cdot SEg1^t\right) \cdot e^{\left(-Sk1 \cdot Sg1 \cdot t\right)}}{\left(Sx1 \cdot e^{\left(-Sk1 \cdot Sg1 \cdot t\right)} + 1\right)^{\frac{1}{Sg1} + 1}}$$

*Figure C: Energy $E_G(t)$ in red projection  vs Exergy $E_X(t) = \alpha \cdot E_G(t)$ where $\alpha = 0.2$ . The blue curve is the natural logarithm of $E_G(t)$. By inspection, the slope (d $\ln(E_G(t))dt$ is greater than 1. The curve of $\alpha \cdot \varepsilon$ is, first of all a flatline because it is a constant. Second it is always less then 1 and $\geq 0$. The equation shown is the function I developed  that fits the growth curve of fossil fuels. This equation is based on industry equations for modeling the output of wells (Richards equation), and is a methane emission model. This is shown strictly as a confirming example. The full equation for fossil fuel consumption into energy is probably what fits here. That one is more complicated.*

However, in figure C we see that the relation $E_a(t) = \alpha \cdot E_G(t)$ is a proportion. In this case (treating $\alpha$ as a constant) $\alpha$ becomes part of the derivative in a nice way. We confirm by inspection of slopes (figure C) that the first derivative of two curves (red and blue) is not identical when it is a proportion.

Therefore, $d \ln(E_a(t))/dt \neq d \ln(E_G(t))/dt = \eta$

From this we know that there are two or three rates, not one, and they are related by $\alpha$. I will assign the derivative of $\ln(E_G(t))$ the value $\eta$ and use subscripts for the others.

$d \ln(E_x(t))/dt = \eta_x$ and $d \ln(E_a(t))/dt = \eta_a$  and  $d \ln(E_G(t))/dt = \eta$
Therefore:
$\eta \equiv d \ln(E_G(t))/dt = \eta(t)$

**The system of equations**

Putting this together, the basic system of equations is:

$E_a(t)$ [J] / $\Lambda(t)$ [J/\$] = $Y(t)$ [\$]
$E_a(t)$ [J] / $Y(t)$ [\$] = $\Lambda(t)$ [J/\$]
$Y(t)$ [\$] · $\Lambda(t)$ [J/\$] = $E_a(t)$ [J]

Thus, it appears it should be possible to create a reasonable $\Lambda(t)$ dataset, and fit functions to that dataset, at least over periods of decades.

**$C(t)$ and $W(t)$ — Beyond this system of equations**

The function called $W(t)$ in Lotka's wheel is called $C(t)$ in Garrett 2011. Here I will use $C(t)$.

$C(t)$ [\$] ≡ $\sum Y(t)$ [\$]

Equations 4, 5, and 6, of Garrett 2011 I cannot make sense of.

**Equation 4 appears to be a typo**

$a = \lambda C(t)$ appears to be a typo. Rewritten, $a = \lambda C(t)$ is
$E_a(t)$ [J] = $\Lambda(t)$ [J/\$] · $\sum Y(t)$ [\$]
The dimensions work, except that this can only be true if we replace $C(t)$ by $P(t)$. Then
$E_a(t)$ [J] = $\Lambda(t)$ [J/\$] · $Y(t)$ [\$]

**Equation 5 doesn't work**

$dC/dt = 1/\lambda\ d\ a(t)/dt = \alpha/\lambda\ d(E_G(t))/dt = \alpha/\lambda \cdot w = \eta/\lambda\ a$   eq. 5 of Garrett 2011
Taking this piecemeal, left to right, inserting transitions.
$dC(t)$**[\$]***/dt* = $Y(t)$**[\$]** =
  ≠ $1/\lambda\ d\ a(t)/dt$  This is not true because there is no derivative needed.
fix it:
$1/\Lambda/(t)$[J/\$] · $E_a(t)$ [J] = $\alpha/\Lambda/(t)$[J/\$] · $E_G(t)$ [J]  There is no derivative in this equality

$\alpha/\Lambda/(t)$[J/\$] · $E_G(t)$ [J] ≠ $\alpha/\Lambda/(t)$[J/\$] · $w$ [J]   Because $w = E_x(t)$ [J] = $\alpha \cdot \varepsilon \cdot E_G(t)$ [J]
fix it:
$\alpha/\Lambda/(t)$[J/\$] · $E_G(t)$ [J] = $\alpha\varepsilon/\Lambda/(t)$[J/\$] · $E_G(t)$ [J]
substitute $\eta$ with $d \ln(\Delta G)/dt$ and an inequality results.
$\alpha\varepsilon/\Lambda/(t)$[J/\$] · $E_G(t)$ [J] ≠ $\underline{d \ln(\Delta G)/dt}/\Lambda/(t)$[J/\$] · $E_a(t)$ [J]
fix it:
$\alpha\varepsilon/\Lambda/(t)$[J/\$] · $E_G(t)$ [J] = $E_x(t)$ /$\Lambda/(t)$[J/\$]   The variable $\eta$ has to be thrown out to work.

**Equation 7 doesn't work**

$$C(t) = \int_0^t P(t)\,dt = \frac{\alpha}{t}\int_0^t w(t)\,dt \qquad\qquad \text{eq. 7}$$

The task here is to transform $P(t)$ into any $w(t)$ form.

I will use $w(t) = E_x(t) = \varepsilon \cdot E_a(t) = \varepsilon \cdot \alpha \cdot E_G(t)$

*So, from above:* $\mathbf{P} \equiv Y(t) = \dfrac{E_x(t)}{\Lambda(t)}$

$$\int_0^t P(t)\,dt = \int_0^t \frac{\varepsilon \cdot \alpha \cdot E_G(t)}{\Lambda(t)}\,dt = \frac{\varepsilon \cdot \alpha}{t} \int_0^t \frac{E_G(t)}{\Lambda(t)}\,dt \neq \frac{\alpha}{t} \int_0^t w(t)\,dt$$

It is not possible to get to the equation 7 formulation right side.

The reason is found in the original note on $w(t)$. (This Garrett 2011 $w(t)$ is not the same as GGK22's $w$.) This $w(t)$ is a conflation of two things into one. The work ($w$) is not the same as the $Y(t)$ result. Instead, $w$ is the exergy consumed to create $Y(t)$, and $\Lambda(t)$ is the transform function.

In addition, a minor issue is that if $w$ is supposed to be the exergy, and we are not actually talking about $Y(t)$, this removes $\Lambda(t)$ and we want to use the exergy, $E_x(t)$, and it's still not possible to get an equality.

**Nolthenius comments – Response to Nolthenius**

Nolthenius displays familiarity with Garrett's antecedent work to GGK22. There is little engagement with the subjects treated in Hanley's Critique. (Hanley2025). There is no mention of the issue with real GDP going to zero. There is no discussion of the graphs and the datasets I worked with.

**The first operative quote:**

> *"I've done an independent investigation into the validity of the GGK22 and prior work on what I've come to call the "Power/Wealth Relation" (P/W); i.e. the constancy of the ratio of the inflation corrected sum total of all human production, and the current consumption rate of primary power. I can confirm that the PW relation is indeed essentially a constant over the period since 1970."*

First, I did not review Nolthenius' slides for my article, I could not compare Nolthenius' data to Garrett's. Second, if one examines my graphs, the time from 1970 forward is in fair agreement with Garrett, although definitely not flat. (See fig. D below) The only thing that is almost flat is the *W* from GGK22's spreadsheet when divided by Exajoules from GGK22's spreadsheet. (Red dashed curve in fig. D below.)

I cannot reconcile the GGK22 supplement W with applying GGK22's summation algorithm on the GDP data of GGK22's spreadsheet.

**Figure: Three graphs compared, Hanley OWID, GGK22 W/E, and GGK22 GDP summation.**

[Figure]

*Figure D In red is the GGK22 W dataset divided by GGK22's Exajoule dataset. In blue is the GGK22 GDP column summation according to GGK22's method for creating W, divided by the GGK22 Exajoule dataset. In black is my GWP dataset independently obtained from Maddison with the method of interpolation, divided by my independently obtained exajoule dataset. These figures are quite close. However I cannot reconcile the red graph with the methods described in GGK22.*

**Second operative quote:**

*"I too find Hanley's contentions just as puzzlingly unsupported and unsupportable. I don't have any disagreements with Garrett's points."*

Based on this review I must presume:

**Nolthenius does not believe dividing by zero** causing inflation to go to infinity is a problem. I say this is nonsense. Nolthenius provides no rationale for how this going to infinity due to dividing by zero could be so.

Does Nolthenius divide by zero on a regular basis in his work? If so, please explain. As an astronomer, he should understand the light-speed gamma limit equation I used as a very rare example of such an equation in the real world.

Might I ask if he saw any evidence that this claim could be true in the empirical data I provided? If so, please advise.

**Nolthenius thinks that an exajoule value in the year 1** that is several times the high confidence value in 1800 is correct. I provide a discussion in my response to Garrett. If he has evidence or rationale to back this up, please advise.
See:  https://doi.org/10.5194/egusphere-2025-699-AC3

**Nolthenius thinks that my graphs showing** gross disparity between GGK22's dataset and my attempt to reproduce it, including an attempt to reproduce it using GGK22's data are wrong in some way. See Fig D above. GGK22 does make the methods quite clear.
Might I ask exactly how Hanley's are wrong?  Is there a place where a conversation can begin? Did Nolthenius perform those summations specified in GGK22 himself, and not find them in error using GGK22's spreadsheet?  If so, I would like to see the spreadsheet.
For that matter, how did he perform the P/W calculations of his own starting in 1970?  Did he use GGK22's W dataset (Column C of GGK22's "Reconstruction" spreadsheet supplement?) One cannot get the result that GGK22 shows in that W (column C) by performing a summation on GGK22's GDP data (Column B of GGK22's "Reconstruction") starting in 1970, nor using any dataset starting in 1970. This is should be impossible because of the necessary shape of the early period of a summation curve.
Perhaps Nolthenius took the GGK22 W dataset at face value? Please advise.

**1. Adding the Shadow Economy:**

The shadow economy is not part of GGK22, nor is the shadow economy mentioned by Hanley. Both Garrett and myself make use of datasets produced by others.  Thus, while this could, perhaps, be brought up as a commentary on Garrett's datasets in general, it is not relevant to this submission. This submission is very specific and focused.
I would refer this shadow economy issue to Maddison project[13], Ian Morris[14], or someone similar who has the depth of knowledge to do something useful with it that is likely to be correct. There are serious issues with "shadow economy" work. For instance, while money changes hands to provide addictive drugs with high mortality rates, is this work productive?  Similarly, are robberies and murders for hire productive? These are thorny issues.

**2. The P/W Constancy over different time scales:**

Hanley2025 is an examination of Garrett's data and algorithms presented in GGK22. That examination shows that the W dataset does not conform to Garrett's claims. See Fig A. See the Hanley critique itself. See my response to Garrett  https://doi.org/10.5194/egusphere-2025-699-AC3 and to Jarvis https://doi.org/10.5194/egusphere-2025-699-AC5 .
GGK22 makes very specific claims going back to the year 1. We are not talking about just after 1970, as Nolthenius appears to be referencing. In my graphs and discussions I make it clear that there is a very interesting low slope starting in 1970. I also make clear that using the summation method specified, the W/E curve (what would be P/W in Nolthenius' parlance) just does not look anything like
* * *
13   Maddison Historical Statistics, https://www.rug.nl/ggdc/historicaldevelopment/maddison/?lang=en
14   Morris, Ian (2013) The Measure of Civilization. Princeton University Press. New Jersey.
     https://kingcenter.stanford.edu/people/ian-morris

what Garrett shows. Period. One cannot just handwave this away. If one is to say, "No, the GGK22 W/E curve is correct," one needs to explain how and why.  Presenting a graph from some slides means nothing here, except as a starting point for examination of why. Because my analysis excludes this being flat.

   If someone is going to tell me I am wrong about the data, then they need to do it using the data of this paper, not some other thing they did without dataset clarity. The data tells the story.

**3. The PPP vs MER argument:**

   This section is incorrect, and this should not be that hard to understand. This is the third time now that I have addressed it. So I will try add a bit to that discussion to clarify.

   Garrett justifies use of MER, and creates an adjustment factor for it, *pi,* in GGK2011. The logic for doing so, quoted below, I do not follow. Essentially, what this is saying is that using trade figures of the global reserve currency, which status permanently tilts the US trade balance to deficit, this gives a more realistic picture? How could that be? The USA has run trade deficits since Bretton-Woods and always since 1976[15]. There is a rough ratio of $1 trillion in US trade deficit for every $160 trillion in GWP.

   I will also point out that the basis of Garrett's work is to move toward an energy basis for currencies. This concept has also been embraced by Ian Morris and others, with the wheat-dollar kcal standard, that relates food to other forms of energy. Both classical energy, and a basic good like wheat in kcal (or rice kcal, etc.) represent a facet of the PPP approach. The persistence of this concept error is hard to fathom.

> *"...because the focus of this study is energy production and associated $CO_2$ emissions, rather than national standard of living, it is historical records of market exchange rate valuations that are used. Exchange rate measures of production P are assumed to most accurately reflect the total energy costs associated with manifesting products and services in the respective nations where they are consumed." pg 453 Garrett 2011[16]*

   We are talking about ancient times to the present, a 2000 year span. For historical and comparative purposes PPP is the way to do it, unless one is talking about current period financial flows. There is no dearth of literature on this. I highly recommend The Measure of Civilization.

   ***Angus Deaton & Alan Heston***[17]. "*Purchasing power parity data provide a common measuring rod that allows comparison, not only of India and America now, but of India now with Britain before the industrial revolution.*"  Emphasis mine. The operative word here is "allows". Without PPP such historical comparisons are inaccurate.

   ***Tim Cullen***[18] speaking to the present period, "*...market-based rates are relevant only for internationally traded goods. Nontraded goods and services tend to be cheaper in low-income than in*

15 Macrotrends U.S. Trade Balance 1970-2025 https://www.macrotrends.net/global-metrics/countries/usa/united-states/trade-balance-deficit
  Federal Reserve Bank of St. Louis (1961) The United States Balance of Payments 1946-1960.
  https://fraser.stlouisfed.org/files/docs/publications/frbslreview/rev_stls_196103.pdf
16 Garrett, T. J. (2011) Are there basic physical constraints on future anthropogenic emissions of carbon dioxide?, Climatic Change, 104, 437–455, https://doi.org/10.1007/s10584-009-9717-9
17 Angus Deaton and Alan Heston, (2010) Understanding PPPs and PPP-Based National Accounts. American Economic Journal: Macroeconomics, vol. 2, no. 4, pp. 1-35  DOI: https://doi.org/10.1257/mac.2.4.1
18 Tim Callen (2007) PPP Versus the Market: Which Weight Matters?  Finance & Development, IMF
  https://www.imf.org/external/pubs/ft/fandd/2007/03/basics.htm

*high-income countries. ...Any analysis that fails to take into account these differences in the prices of nontraded goods across countries will underestimate the purchasing power of consumers in emerging market and developing countries and, consequently, their overall welfare.*"

"*Drawbacks of PPP. The biggest one is that PPP is harder to measure than market-based rates. The ICP is a huge statistical undertaking.*"

**IPCC discussion of MER vs PPP**[19].

*"On the question of whether PPP or MER should be employed in economic scenarios, the general recommendations are to use PPP where practical.[3] This is certainly necessary when comparisons of income levels across regions are of concern."*

To capsulize the problem again, it is nonsensical to talk about MER in the year 1000 or the year 1 CE. Why? Because dollars did not exist then. There are no "exchange rate records" to use. Yes, GGK22 created a *pi* value used to adjust to MER back to the year 1. This *pi* value, while I do understand its derivation, is meaningless.

In any case it does not make a meaningful difference for the purposes of this discussion. I show this in my reply to Garrett

See: https://doi.org/10.5194/egusphere-2025-699-AC3

**4. Inflation measure:**

The replication GWP data comes from Maddison project. Maddison puts far more care and sophistication put into this than Nolthenius displays here, and nobody should expect anything else. To expect it would be like expecting an undergrad in atmospheric science to get everything right. I chose Maddison project/OWID for that reason. These people are the experts. From Blair Fix' recommendation, I add Ian Morris and colleagues to this. Maddison's founding of this field is half a century and more ago. We should build on them to avoid straying into crank territory by accident.

Further, the graphs I provide using Garrett's data show that this inflation measure does not really matter for the points being made.

See: https://doi.org/10.5194/egusphere-2025-699-AC3

**5. Civilization Obeys Thermodynamic laws:**

I agree. I said that Garrett's 2011 statement – that if $\lambda$ is not constant, the thermodynamic model is false – is not correct. See my reply to Jarvis https://doi.org/10.5194/egusphere-2025-699-AC5 .

The Yakovenko work in econophysics is interesting, but it is a different model than the one created by Garrett. The Yakovenko work correlates with, and broadly supports Ian Morris' work that is well discussed in his book, "The Measure of Civilization." Morris provides a detailed, nuanced discussion of the archeological considerations that go into estimation of energy consumption (which he calls energy capture), and by extension, GDP (to some extent) back to 14,000 BCE, a span of 16000 years. This book is well worth reading.
* * *
19   3.2.1.4 The use of MER in economic and emissions scenarios modelling.  Climate Change 2007: Working Group III: Mitigation of Climate Change. https://archive.ipcc.ch/publications_and_data/ar4/wg3/en/ch3s3-2-1-4.html

Shinckus[20] provides a good overview of Econophysics in his 2018 doctoral thesis, "When Physics Became Undisciplined An Essay on Econophysics." He suggests that the field originated from physics finding finite solutions of stable Levy processes[21], and applying this to research problems abandoned by finance in the 1960's.

**6. Thermodynamics and Human Agency:**

I have little argument with the generic comments on neoclassical economics and its antipathy to thermodynamics. I might characterize antipathy as incomprehension instead.

This is beside the point for the article under review. My interest is to make Garrett's thermodynamic approach more accessible and to validate the Lotka's Wheel paper datasets and claims. I do not want to throw the baby out with the bathwater.

**Nolthenius' disagreements with Garrett:**

Here Nolthenius discusses Garrett's broader work, which I touch on very lightly in the introduction, and have a basic discussion of in my reply to Jarvis.
See: https://doi.org/10.5194/egusphere-2025-699-AC5 .

Nolthenius' points are eloquently put, however, outside of scope for this article. I will make some comments anyway.

**A. Examining Nolthenius' argument about interest rates and money creation**

This private sector mechanism of money creation through loans (which is secondary to the deficit spending that creates the net positive monetary equity for the private sector) is not necessarily constrained as suggested. Historical data shows that the Fed-funds-rate can be high in an economy making lots of loans. https://fred.stlouisfed.org/series/FEDFUNDS

Yes, there are periods of recession somewhat correlated with rise in Fed-funds-rate, but the bulk of time the Fed-funds-rate was quite high by post 2010 standards. One should correct the Fed-funds-rate using CPI, and when this is done, the picture becomes quite interesting. Businesses will borrow when rates are high, if inflation is also high. What the *net* real interest rate is on borrowing – that is the question. This is similar to an issue I talked about with my original economics mentor, Kevin Walsh regarding real interest rates in the Islamic empires over the centuries of conquests, when the regime was formally zero interest. This could have become an area of study, but it was superseded by the 2008 crisis.

**B. The belief that money creation causes inflation**

This a well debunked neoclassical idea that comes from Friedman's "Helicopter money" hypothesis (Keen 2021, pg 19)[22]. Yes, having more money in the hands of people that will spend

20   Schinckus, Christophe (2018) When Physics Became Undisciplined An Essay on Econophysics. Girton College, University of Cambridge. DOI: https://doi.org/10.17863/CAM.27052
21   https://en.wikipedia.org/wiki/L%C3%A9vy_process
     *Raible, S. (2000). Lévy Processes in Finance: Theory, Numerics, and Empirical Facts. https://d-nb.info/961285192/34*
22   Steve Keen (2021) The New Economics: A Manifesto.  Polity Press. ISBN  9781509545308, https://books.google.com/books?id=KEVOEAAAQBAJ

it allows price increases, but the cause of the 70's inflation was oil shocks (increased energy price) and the cause of today's inflation is supply side recovery price increases and transport constraint. There has also been a natural origin biological onslaught on chickens which has drastically cut the food supply related to birds. Chicken and eggs have been a cheap staple and suppliers have had difficulty rebuilding flocks.

Businesses kept operating at a loss during the COVID lockdown. Ships that cost millions per month to maintain sitting in port were triaged by age and future mainenance costs, and scrapped to raise cash and lower current expenses. So supply was constrained by shipping capacity, causing supply not to meet demand. Few corporations had the foresight and funds to charter or lease ocean freighters to prep for recovery[23]. Businesses also needed to make up their losses to a reasonable extent to stay solvent, causing upward pressure on pricing.

The money created by the Fed (QE) did not go into the hands of workers, nor did cutting interest rates. QE increased the money supply, slowing money velocity, and supporting stock prices. The project of stimulating loans by lower interest rates is "pushing a rope" to force demand.

The idea that when energy costs go up, other prices go down is false. Higher energy costs get passed along in pricing, because energy is fundamental to everything in civilization. The 70's OPEC oil shocks did not lower prices, but did cause inflation. Goods may become more scarce because of high energy price. This *may* result in fewer sales, and it *may* result in substitution of cheaper goods if such are available. It will not drive prices in general down. That is backwards.

**C. I would agree that ascribing all of recession to energy is not correct**

However, I don't think I saw that claim in Garrett's earlier work. If it is there, I apologize.

To see the EROEI signal one would need to find ways to remove the "fools recessions" from the record somehow, or at least look at long time periods. Until it is done I don't think an empirical test of Garrett's hypothesis is possible. I do think Garret's hypothesis is interesting.

Given that money and finance are a layer within economics that neoclassicals avoid like a plague of locusts, I think such an effort will not occur from neoclassical economists any time soon. After studying macroeconomics and the sometimes weird mathematics in the area, I think that most economists prefer the ivory-tower models without reference to money because the monetary layer is difficult to work with.

**D. In regard to the closing:**

"the validity of the P/W Relation's validity in the data, which was the subject of Hanley's critique"
My critique was very specific. I stepped through GGK22 and found that data did not behave as Garrett said it does. I made observations that I do not see refuted, nor even referenced here, except perhaps by implication. I discussed issues with the claim that real GDP could go to zero because inflation went to infinity.
* * *
23  Alejandra Carranza (2021) Chartered ships emerged in 2021 as a way to get around congestion. Supply Chain Dive. https://www.supplychaindive.com/news/2021-businesses-charter-ships-supply-chain-delays/611563/